# Evaluating 17 methods incorporating biological function with GWAS summary statistics to accelerate discovery demonstrates a tradeoff between high sensitivity and high positive predictive value

Amy Moore [1✉], Jesse A. Marks [1], Bryan C. Quach [1], Yuelong Guo[2], Laura J. Bierut [3], Nathan C. Gaddis[1], Dana B. Hancock [1], Grier P. Page [1,4] & Eric O. Johnson [1,4✉]

Where sufficiently large genome-wide association study (GWAS) samples are not currently available or feasible, methods that leverage increasing knowledge of the biological function of variants may illuminate discoveries without increasing sample size. We comprehensively evaluated 17 functional weighting methods for identifying novel associations. We assessed the performance of these methods using published results from multiple GWAS waves across each of five complex traits. Although no method achieved both high sensitivity and positive predictive value (PPV) for any trait, a subset of methods utilizing pleiotropy and expression quantitative trait loci nominated variants with high PPV (>75%) for multiple traits. Application of functionally weighting methods to enhance GWAS power for locus discovery is unlikely to circumvent the need for larger sample sizes in truly underpowered GWAS, but these results suggest that applying functional weighting to GWAS can accurately nominate additional novel loci from available samples for follow-up studies.

[1] Genomics and Translational Research Center, RTI International, Research Triangle Park, NC 27709, USA. [2] GeneCentric Therapeutics, Inc., Cary, NC, USA. [3] Department of Psychiatry, Washington University School of Medicine, St. Louis, MO, USA. [4] Fellow Program, RTI International, Research Triangle Park, NC 27709, USA. ✉email: almoore@rti.org; ejohnson@rti.org

The genome-wide association study (GWAS) has been widely successful for discovering genetic loci contributing to complex traits[1]. Yet, a survey of the GWAS catalog identified 88 traits without genome-wide significant findings despite theoretically adequate sample size[2]. Traits with worse than expected performance even when thousands of cases are available include autism spectrum disorder[3], heart failure[4,5], major depressive disorder (MDD)[6,7], and some addictions[8-11]. Increasing sample size to increase statistical power for discovery is not always practical, as encountered for rare diseases[12], expensive phenotyping[13], phenotypic heterogeneity[14], hard-to-reach or socially disadvantaged populations[15], and population isolates[16]. Our ability to discover trait-associated loci that are ancestry-specific or subject to gene-environment interaction lags in a field where the overwhelming majority of GWAS samples are of European ancestry[17]. Further, increasing sample size sometimes fails to achieve the expected gain in significant loci[18].

Attempts to improve the discovery power of GWAS without increasing sample size by incorporating functional information, defined here as regulatory annotation of variants or evidence of pleiotropy, is not new[19]. An evaluation of gene- and pathway-based GWAS methods found low sensitivity overall for discovery, and that high sensitivity was achieved at the expense of more false positives[20]. Methods to combine GWAS summary statistics with additional information to perform in silico functional follow-up are plentiful[21-25] and range from fine-mapping to determining the biological underpinnings of the variant-trait association; many of these in silico approaches to uncover the functional causes underpinning GWAS associations have been previously reviewed[20,26-30]. Some authors suggest that a secondary usage of these methods is to augment the ability of a given GWAS to identify novel trait-associated loci. Evaluation of the performance of such methods for locus discovery has been done ad hoc for select methods,[21-23,31] but to our knowledge, a comprehensive evaluation of many methods and multiple GWAS traits against objective criteria has not been published.

To identify suitable method(s) for improving GWAS statistical power to uncover novel loci, we performed the largest, most comprehensive evaluation of published functional weighting methods to date: 17 functional weighting methods, and an unweighted suggestive p-value threshold, applied to multiple waves of GWAS for five diseases and traits. We applied these methods to publicly available GWAS summary statistics and evaluated their ability to nominate trait-associated loci that were confirmed by a subsequently larger, more powerful GWAS, henceforth referred to as GWAS1/GWAS2/GWAS3, for the same trait. To represent varying genetic architectures, phenotypic heterogeneity, and gene regulation by tissue type, we selected three psychiatric traits: schizophrenia, bipolar disorder, and MDD available from the Psychiatric Genomics Consortium (PGC); and two blood cell traits: mean platelet volume (MPV) and white blood cell (WBC) counts, available from the UK Biobank.

## Results

We selected 17 published functional weighting methods; we also evaluated a suggestive p-value threshold of $1 \times 10^{-5}$ as an 18th method (Table 1 and Supplementary Data 1). We applied these methods to five model traits, described in Supplementary Data 2. Nine methods provided results for individual variants, and nine provided gene-based results aggregated across variants. When evaluated on a per-variant basis according to the schema in Supplementary Data 3, the number of nominated variants, after excluding statistically significant variants from GWAS1, ranged from zero to 177,698 in the blood cell traits and zero to 4147 in

the psychiatric traits (Supplementary Data 4 and 5). Statements in these published papers that indicating that the method may increase variant-trait association discovery are provided in Supplementary Note 1.

Briefly, we applied each published functional weighting method (Table 1) to genome-wide summary statistics from each GWAS1 study. Details of additional annotation datasets and statistical significance thresholds used for each method are described in the "Methods" and Supplementary Data 1. To facilitate cross-method comparisons, our primary way to evaluate both variant-based and gene-based method performance used a +/−500 kb window to define a locus, unless specified otherwise. Overlapping loci were merged. To exclude the possibility of methods re-discovering loci already identified as trait-associated in GWAS1, we did not consider loci if they overlapped with a +/−500 kb window surrounding the top variant of a locus that was genome-wide significant in GWAS1. Each functionally weighted GWAS1 was then compared to the corresponding GWAS2 of that trait to identify nominated loci from GWAS1 that overlapped with genome-wide significant loci first identified in GWAS2. A minimum overlap of 250 kb was required. Our scheme for defining classification metrics (True Positive [TP], etc.), is illustrated in Supplementary Data 3. Our primary evaluation metrics, Positive Predictive Value (PPV) and Sensitivity (SN), were derived from these classification metrics.

**Global evaluation.** No method had both high SN and PPV (>0.50, Fig. 1, Quadrant I). In general, there was an inverse relationship between SN and PPV (Fig. 1). Quadrant IV, with high SN and low PPV, was dominated by methods providing variant-level results and by the blood cell traits MPV and WBC. Quadrant II, with low SN and high PPV, was dominated by eQTL-based methods, which tended to nominate fewer loci than the variant-level methods (Supplementary Data 5). Exceptions to the pattern of finding eQTL-level methods in Quadrant II were MTAG and the weighted eQTL methods. These methods nominated fewer loci for their respective traits than was typical for other variant-based methods (Supplementary Data 4).

Quadrant III of Fig. 1, representing low SN and low PPV, included results from all five traits and a preponderance of MDD, specifically around SN = 0 and PPV = 0. Only five out of nine methods nominated any variants for MDD (Supplementary Data 4), which had no significant hits in GWAS1. Like the variant-based methods, only four out of nine gene-based methods yielded any nominations for MDD, and none of those overlapped with the GWAS2 hits for MDD, regardless of the evaluation method used (Supplementary Data 5).

We provide representative Manhattan plots[32] to illustrate the performance of two functional weighting methods for the high PPV (MTAG, Fig. 2a, b) and high SN (LSMM, Fig. 2c, d) scenarios, respectively. When comparing the variants nominated by MTAG for SCZ1, using BPD1 as the pleiotropic trait, relative to both waves of the SCZ GWAS, the nominated variants of MTAG clustered around established "peaks", including some that are just below the genome-wide significance threshold in GWAS1 (Fig. 2a). Some of these variants (e.g., see Chromosomes 3 and 12) are in loci that become significant in GWAS2 (Fig. 2b), contributing to the high PPV of this method-trait combination, while others fall below even the suggestive threshold in GWAS2 (e.g., see Chromosome 7). However, these particular non-significant nominated variants are within 500 kb of the GWAS2 top hit (Supplementary Data 4).

LSMM with global FDR nominated 3395 more variants for SCZ than MTAG, resulting in high SN (Supplementary Data 4). In contrast to MTAG, a striking proportion of these nominated

**Table 1 Description of functional weighting methods.**

| Method name | Classification | Level | Citation | Significance threshold |
|---|---|---|---|---|
| Suggestive | NA | Variant | NA | $p < 1e-5$ |
| GenoCanyon10K | Annotation | Variant | Lu et al. Sci. Rep. 2015 | Prediction Score >0.5 |
| GenoSkyline | Annotation | Variant | Lu et al. PLoS Genet. 2017 | Prediction Score >0.5 |
| Sveinbjornsson | Annotation | Variant | Sveinbjornsson et al. Nat. Genet. 2016 | Annotation-based threshold |
| LSMM | Annotation | Variant | Ming et al. Bioinformatics 2018 | FDR < 0.05 |
| GPA | Pleiotropy | Variant | Chung et al. PLoS Genet. 2014 | FDR < 0.05 |
| MTAG | Pleiotropy | Variant | Turley et al. Nat. Genet. 2018 | $p < 5e-8$ |
| fGWAS | eQTL | Variant | Pickrell. AJHG 2014 | PPA > 0.9 |
| Weighted eQTL | eQTL | Variant | Li et al. Front. Genet. 2013 | $p < 5e-8$ |
| COLOC | eQTL | eQTL | Giambartolomei et al. PLoS Genet. 2014 | Approximate Bayes Factor >0.75 |
| MOLOC | eQTL | eQTL | Giambartolomei et al. Bioinformatics 2018 | Posterior Probability >0.80 |
| Jepeg | eQTL | eQTL | Lee et al. Bioinformatics 2014 | Bonferroni-adjusted Jepeg p-value |
| Sherlock | eQTL | eQTL | He et al. AJHG 2013 | Log Bayes Factor >= 4.0 |
| SMR | eQTL | eQTL | Zhu et al. Nat. Genet. 2016 | FDR q-value < 0.05; Heidi p-value < 0.05 |
| TWAS/FUSION | eQTL | eQTL | Gusev et al. Nat. Gen. 2016 | Bonferroni-adjusted TWAS p-value |
| fastENLOC | eQTL | eQTL | Wen et al. PLoS Genet. 2017 | RCP >= 0.50 |
| EUGENE | eQTL | eQTL | Ferreira et al. JACI 2017 | p-value corresponding to largest FDR < 0.05 |
| UTMOST | eQTL | eQTL | Hu et al. Nat. Genet. 2019 | Bonferroni-adjusted UTMOST p-value |

*FDR* false discovery rate.

variants exhibited a sharp decrease in significance from GWAS1 (Fig. 2c) to GWAS2 (Fig. 2d), contributing to the low variant-based PPV under both FDR options for LSMM; the PPV also remained low in the locus-based evaluation (Supplementary Data 4).

Figure 3 illustrates the performance of gene-based methods. To provide parity in evaluating nominated genes, we calculated gene-based p-values using a modification of MAGMA (see "Methods"). The gene-based methods nominated fewer loci than the variant-based methods. For both EUGENE and SMR, which were applied using Brain eMETA cohort annotations, nominated genes tended to have higher MAGMA p-values (Fig. 3a, c) but lower p-values in GWAS2 (Fig. 3b, d).

**Top method for positive predictive value**. Focusing on the ability of methods to accurately nominate loci that were truly trait-associated but inadequately powered for detection in GWAS1, we compared PPV across all traits (Table 2). When multiple databases were applied to a functional weighting method, we chose its highest PPV to carry forward for overall evaluation. Any method ties were all assigned the lowest rank, and methods that failed to nominate any variants/eQTL/genes were ranked lower (NA) than methods with a PPV of 0%. Overall, the best-performing method was MTAG[33], even after a sensitivity analysis excluding the MDD rankings, where MTAG tied with the weighted eQTL method of Li et al. This ranking was made despite MTAG failing to nominate any variants for MPV (Supplementary Data 4). The best-performing method for MPV alone was Sherlock, which was a middle performer for the three PCG traits.

**Effect of evaluation strategy**. Only one evaluation strategy allowed for direct comparison between variant-level and eQTL-level methods; however, we also considered several strategies that permitted us to rank variant-level and eQTL-level methods amongst themselves, respectively. Among variant-based method evaluations, the top method for SN changes from LSMM to a tie between LSMM and GPA when evaluated on a variant-to-variant comparison; GPA was ranked second by the 500 kb comparison. There were no changes in the top performing method for PPV. Among the eQTL-based method evaluations, there are no changes across the three ways to calculate SN (Supplementary Data 6). For PPV, both the 500 kb and ENSG-boundary-based

comparisons have COLOC as the top performer; however, using Magma to evaluate performance, the top performer is JEPEG, which is otherwise in the middle of the pack for PPV.

**Consistency of true associations nominated across methods**. We evaluated whether loci nominated by multiple methods are more likely to be TP, as running the same summary statistics through multiple methods is cheaper than conducting a larger GWAS. In general, this was an effective strategy. For example, seven methods was the minimum number necessary to achieve PPV ≥ 50% (Supplementary Data 7) for four out of the five traits. For MPV and WBC, we did not see a monotonic increase in PPV with larger numbers of nominating methods, and for MDD, only two methods successfully nominated any TP loci. We examined combinations of functional weighting methods to determine if there existed an ensemble set that consistently achieved PPV ≥ 50% across traits (Supplementary Fig. 4). Across SCZ, BPD, MPV, and WBC, the methods GenoCanyon and LSMM were common to all method ensembles with a minimum of seven methods; however, the inclusion of one or both of these methods does not preclude a false positive (FP). None of the ensemble sets could be used to reliably nominate TPs across traits.

**Evaluating false positives**. Some loci nominated by the functional weighting methods and labeled as FP by our definition may be truly associated with the trait but remain undiscovered in GWAS2. As a sensitivity analysis, we used GWAS3 waves and calculated the PPV of the nominated loci after removing findings from GWAS1, similar to our primary analysis approach. Figure 4 shows the SN and PPV of the functional weighting methods for the three psychiatric traits based on their GWAS3 waves. GWAS3 were not available for the blood cell traits MPV and WBC. Like Fig. 1, no methods appeared in Quadrant I. In general, PPV was higher and SN was lower when using GWAS3, compared to using GWAS2, as the gold standard. A substantial number of the method-trait combinations remained in Quadrant III with low SN and PPV. Supplementary Data 8 shows that no methods had a worse PPV when GWAS3 was used as the gold standard rather than GWAS2. An improved PPV when compared to the larger GWAS3 is expected when additional nominated loci are trait-associated. For BPD and MDD, most methods with any successful nominations still had PPV < 50% when compared to

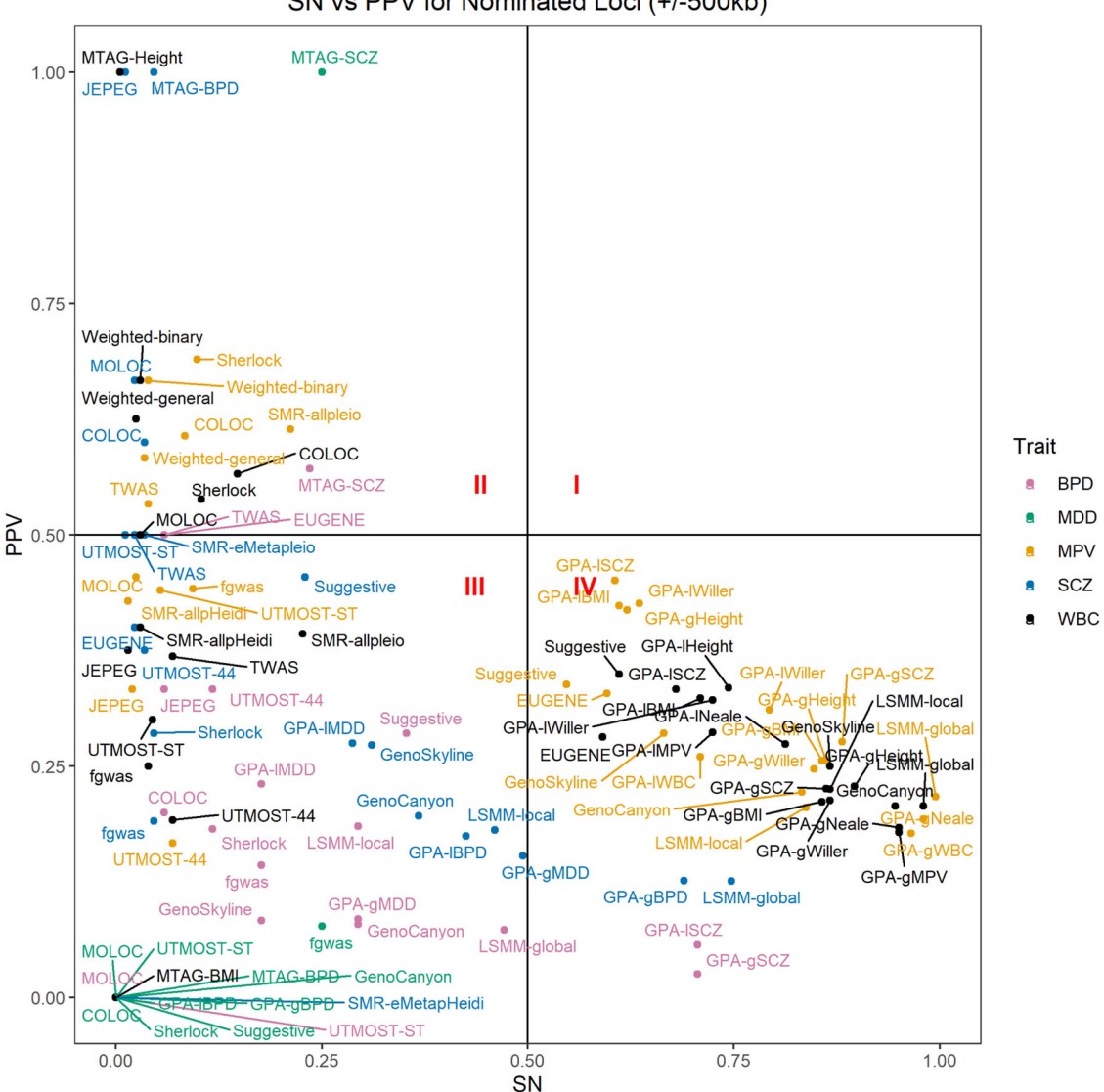

**Fig. 1 Relationship between sensitivity and positive predictive value for all method-trait combinations.** Scatterplot of the relationship between sensitivity (SN; proportion of loci that are significant exclusively in the second wave GWAS [GWAS2] that are also nominated by a given method when applied to GWAS1) and positive predictive value (PPV; proportion of all nominated loci by a given method when applied to GWAS1 that are also significant exclusively in GWAS2) for all method-trait combinations whose results contained at least one gene or locus that was nominated as trait-associated by each method, respectively, after excluding loci identified in GWAS1. SN and PPV were calculated using the +/−500 kb locus-based evaluation and requiring a minimum overlap of 250 kb between nominated loci and GWAS2 significant loci. Horizontal and vertical lines denote PPV and SN of 50%, respectively.

GWAS3. For the variant-based methods, only MTAG out-performed the approach of simply using a suggestive *p*-value threshold in the original GWAS1 when using either GWAS2 or GWAS3 as the gold standard for both SCZ and BPD (Supplementary Data 8).

**Evaluating the stringency of genome-wide significance**. We present a subset of the SN versus PPV results shown in Fig. 1 as the SN versus PPV results for only PGC traits in Supplementary Fig. 1 for ease of comparison with sensitivity analyses. The evaluated methods do not employ a consistent strategy for multiple testing correction or determination of statistical significance. We used a Bonferroni correction based on the number of valid test statistics for methods that calculated a *p*-value but did not provide a prespecified significance threshold. To evaluate whether this conservative approach hampered our ability to detect trait-

associated loci, we performed a sensitivity analysis by calculating a local FDR and using a *q*-value of 0.05 as the threshold for statistical significance for those methods previously subjected to a Bonferroni correction. Results were largely unchanged (Supplementary Fig. 2), except for a substantial drop for MTAG and JEPEG, which had achieved perfect PPV with some traits when using the Bonferroni correction.

**Evaluating the amount of overlap**. We evaluated the impact of our primary choice for defining a minimum overlap (250 kb) between nominated loci and gold standard loci. We performed a sensitivity analysis utilizing different minimum overlaps of one base, 500 kb, and 750 kb. In general, we found a slight reduction in SN and PPV with increasing size of the required overlap for all five traits (Supplementary Fig. 3a–e). However, we did not find

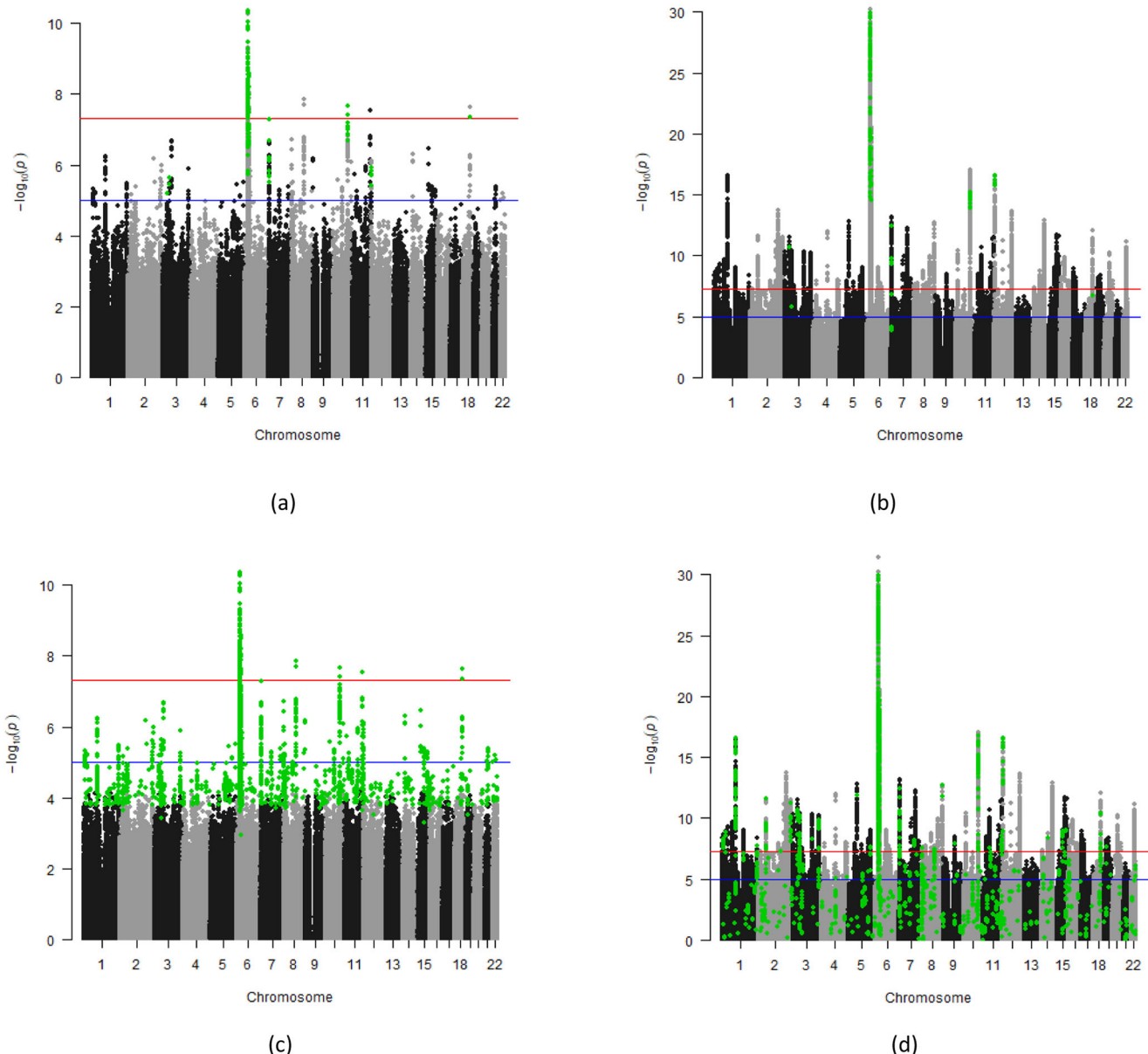

**Fig. 2 Identifying variants nominated by MTAG and LSMM methods in wave 1 schizophrenia GWAS results and their results in the wave 2 GWAS results.** Manhattan plots of schizophrenia GWAS waves 1 (SCZ1; **a**, **c**) and 2(SCZ2; **b**, **d**) with the variants nominated by MTAG (**a**, **b**) using bipolar disorder GWAS wave 1 (BPD1) as the pleiotropic trait and the LSMM method using a global FDR (LSMM) (**c**, **d**) highlighted in green. *P*-values were derived from the publicly available downloads of SCZ1 and SCZ2 provided by the Psychiatric Genomics Consortium, respectively. These plots include the full downloadable GWAS summary statistics for both SCZ GWAS waves, without excluding significant GWAS1 regions.

that our results, particularly our high PPV method-trait combinations, were dependent on overlap size.

## Discussion
Our comprehensive, multi-method evaluation presents scenarios where functional weighting methods might prove helpful in expanding the number of novel loci uncovered by GWAS in lieu of increased sample size. None of the eighteen methods achieved both high PPV and high SN, which would have been the ideal result: nominating a substantial proportion of TP loci that would be found in the next GWAS wave without nominating excessive FP loci. Instead, our evaluation demonstrated that the use of functional weighting methods presents a tradeoff between high SN and high PPV. MTAG[33] had the best performance overall

with respect to PPV, and LSMM with respect to SN. To weight equally SN and PPV by choosing the method that yields the highest F1 score, or harmonic mean of SN and PPV, when evaluating the +/−500 kb locus overlap, one can simply use the common suggestive *p*-value threshold of $1 \times 10^{-5}$.

When comparing functional weighting GWAS results to standard GWAS results from larger sample sizes as the gold standard, the PPV for many method-trait combinations exceeded 50%, indicating that most nominations were trait-associated by the standard defined here. For BPD and SCZ, where GWAS1 were adequately powered to detect genome-wide significant associations, most eQTL-based methods were able to consistently nominate TP loci when compared to GWAS3 as the gold standard; however, SN decreased across method-trait combinations, indicating that functional weighting GWAS methods combined

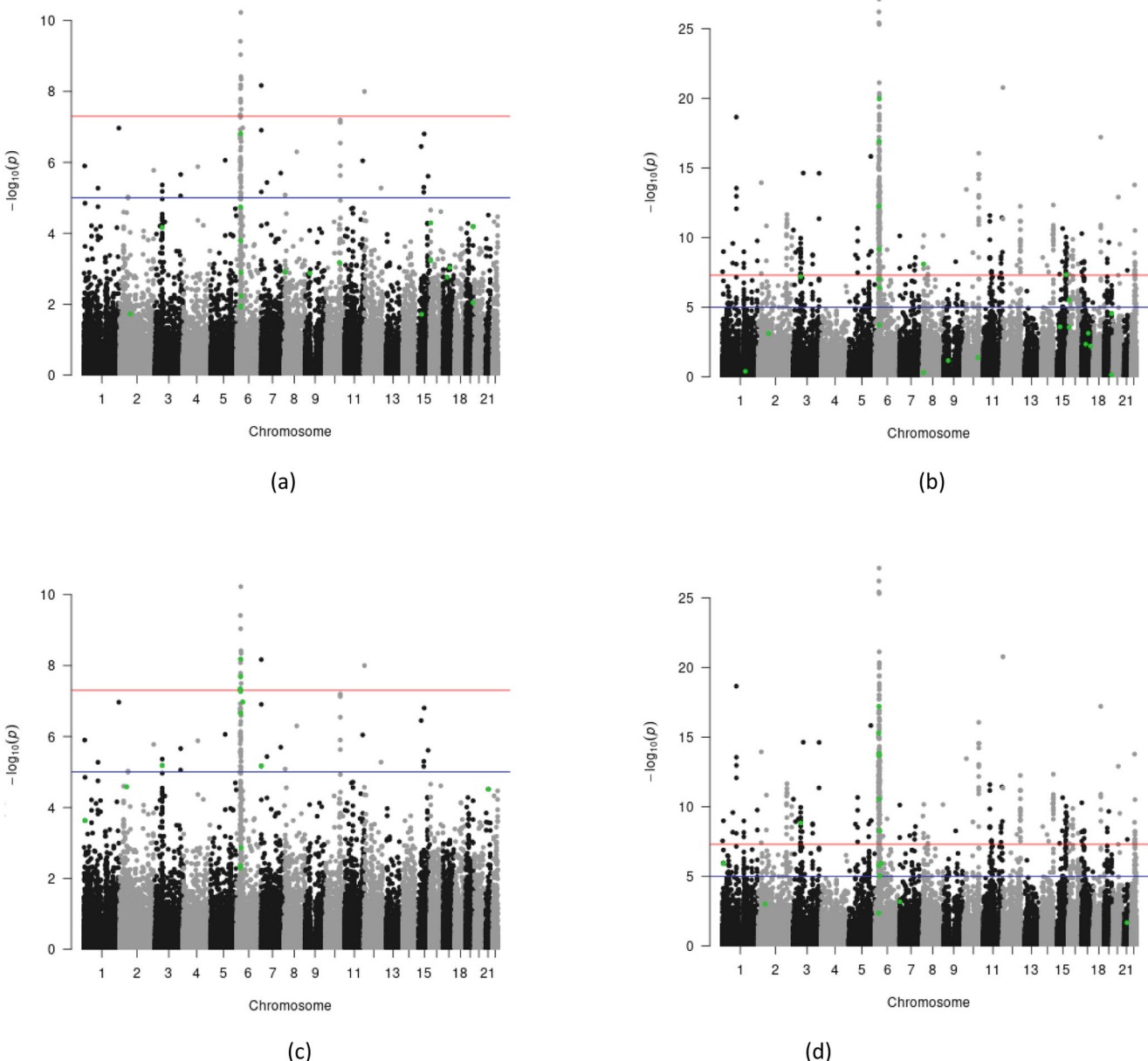

**Fig. 3 Identifying variants nominated by EUGENE and SMR methods in wave 1 schizophrenia GWAS results and their results in the wave 2 GWAS results.** Manhattan plots of schizophrenia GWAS waves 1 (SCZ1; **a**, **c**) and 2 (SCZ2; **b**, **d**) with the variants nominated by the EUGENE (**a**, **b**) and SMR (**c**, **d**) methods using Brain eMETA cohort annotations (SMR2) highlighted in green. *P*-values were derived from applying MAGMA to SCZ1 and SCZ2, respectively. These plots include the full downloadable GWAS summary statistics for both SCZ GWAS waves, without excluding significant GWAS1 regions.

with contemporaneous annotation databases were unable to identify a correspondingly large fraction of the trait-associated variants that can be captured with a larger GWAS sample size incorporating tens of thousands of additional cases. For SCZ, functionally weighted GWAS applied to SCZ1 uncovered 74.7% of the loci that were genome-wide significant for the first time in SCZ2, which dropped to 58.1% of the loci that were genome-wide significant for the first time in SCZ3. Differences in the number of methods that were able to nominate any loci across the three psychiatric traits reiterate that trait heterogeneity, frequency, and variant association magnitude contribute in combination with study sample size to determine the minimum adequate GWAS[34].

If the goal is high-confidence nominations, then there is little additional cost to applying multiple functional weighting methods using publicly available annotation data. Our findings did not show an ideal ensemble approach, whereby nominations that

intersected a subset of specific methods subsequently became genome-wide significant in later GWAS waves. Instead, for traits that had relatively few genome-wide significant loci identified in GWAS1, we found that increasing the number of functional weighting methods increased the PPV of those nominations. An ensemble approach may be achievable in the future as functional annotation data in disease-relevant tissues and cell types expands, enabling a comparison of methods with more complete annotations across the genome.

Applying functional weighting methods for the discovery of novel loci and variants carries important considerations. First, applying these methods to a GWAS in the "dead zone" of statistical power[18], where no genome-wide significant loci have been identified using standard GWAS, may not provide a reliable approach to find trait-associated variants. Using MDD as an example, only a minority of the tested methods were able to

**Table 2 Ranking of all methods by best performing PPV, as measured by locus (+/−500 kb).**

| Method | BPD | | MDD | | SCZ | | MPV | | WBC | | Median Rank | Median Rank excluding MDD |
|---|---|---|---|---|---|---|---|---|---|---|---|---|
| | Best PPV | Rank | Best PPV | Rank | Best PPV | Rank | Best PPV | Rank | Best PPV | Rank | | |
| MTAG | 0.571 | 1 | 1.000 | 1 | 1.000 | 2 | NA | 18 | 1.000 | 1 | 1 | 2 |
| TWAS/FUSION | 0.500 | 3 | NA | 18 | 0.500 | 6 | 0.533 | 4 | 0.368 | 8 | 6 | 5 |
| UTMOST- (best result) | 0.333 | 5 | 0.000 | 9 | 0.500 | 6 | 0.440 | 8 | 0.300 | 11 | 8 | 7 |
| Jepeg | 0.333 | 5 | NA | 18 | 1.000 | 2 | 0.333 | 11 | 0.375 | 7 | 7 | 5 |
| COLOC | 0.200 | 8 | 0.000 | 9 | 0.600 | 4 | 0.607 | 3 | 0.566 | 3 | 4 | 3 |
| EUGENE | 0.500 | 3 | NA | 18 | 0.400 | 8 | 0.329 | 12 | 0.282 | 12 | 12 | 10 |
| Sherlock | 0.182 | 11 | 0.000 | 9 | 0.286 | 9 | 0.690 | 1 | 0.538 | 4 | 9 | 3 |
| moloc | 0.000 | 14 | 0.000 | 9 | 0.667 | 3 | 0.455 | 5 | 0.500 | 5 | 5 | 4 |
| GPA | 0.231 | 7 | 0.000 | 9 | 0.275 | 10 | 0.451 | 6 | 0.335 | 10 | 9 | 8 |
| Suggestive | 0.286 | 6 | 0.000 | 9 | 0.455 | 7 | 0.338 | 10 | 0.349 | 9 | 9 | 8 |
| fgwas | 0.143 | 10 | 0.077 | 2 | 0.190 | 13 | 0.442 | 7 | 0.250 | 14 | 10 | 10 |
| GenoSkyline | 0.083 | 12 | NA | 18 | 0.273 | 11 | 0.285 | 13 | 0.250 | 14 | 13 | 12 |
| GenoCanyon | 0.079 | 13 | 0.000 | 9 | 0.196 | 12 | 0.222 | 14 | 0.207 | 16 | 13 | 13 |
| LSMM | 0.185 | 9 | NA | 18 | 0.181 | 14 | 0.216 | 15 | 0.225 | 15 | 15 | 15 |
| SMR (best performing) | NA | 18 | NA | 18 | 0.000 | 15 | 0.429 | 9 | 0.400 | 6 | 15 | 9 |
| Weighted eQTL | NA | 18 | NA | 18 | NA | 18 | 0.667 | 2 | 0.667 | 2 | 18 | 2 |
| fastENLOC | NA | 18 | NA | 18 | NA | 18 | NA | 18 | NA | 18 | 18 | 18 |
| Sveinbjornsson | NA | 18 | NA | 18 | NA | 18 | NA | 18 | NA | 18 | 18 | 18 |

Ties between methods resolved using the Olympic method.
BPD bipolar disorder, MDD major depressive disorder, SCZ schizophrenia, MPV mean platelet volume, WBC white blood cell count, PPV positive predictive value, kb kilobase, NA not applicable.

nominate any loci for MDD, and few nominated variants were significant in MDD2 (Supplementary Data 4, 5) or MDD3 (Supplementary Data 8). This difficulty in nominating TP loci for MDD suggests that functional annotation is unlikely to overcome insufficient statistical power for GWAS with sample sizes that are far below what is needed to identify robust genome-wide significant loci. For these situations, increasing the GWAS sample size is ideal[6,35,36]. However, if a second, contemporaneous GWAS of a highly pleiotropic trait is available, applying pleiotropy-based methods such as MTAG or GPA may provide an alternative approach. Although identifying the minimum required SNP-based genetic correlation is beyond the scope of this analysis, we note that SCZ and MDD have a SNP genetic correlation ranging from 0.34–0.51, depending on the study[6,37–39]. It is also worth considering that while improvements in trans-ethnic GWAS methods boost discovery power[40], uncovering ancestry-specific loci will require investments to increase the sample size of either the ancestry-specific GWAS or the ancestry-specific functional database[41,42].

Second, by using a distance-based locus definition, we could not evaluate whether the nomination captured the putative causal variant or gene identified in GWAS2. For example, MTAG and fgwas successfully nominated the HLA region as associated with MDD. As is typical with findings located in this region, more work is necessary to identify the causal mechanism for the association between HLA and MDD; initial work by the PGC noted that the C4A and C4B genes were unlikely to be causal for MDD[6], although these genes were functionally characterized as potentially causal for SCZ[43]. Subsequent fine-mapping of the classical MHC region by the PGC also did not support variation in C4 genes to be the source of the MDD association[44], though an eQTL-based analysis identified C4A as a candidate risk gene for MDD[45]. The extended HLA region was confirmed in a subsequent GWAS of MDD, though with a different lead SNP[35]. In our study, MTAG used pleiotropy to find what is likely a true association between the extended HLA region and MDD earlier than it could be discovered with the MDD GWAS1 sample size,

but this is likely driven by linkage disequilibrium in the region, rather than genuinely shared pleiotropy between causal genes[44].

Third, our study focused on comparing three categories of methods which we describe as annotation, pleiotropy, and eQTL. These three categories represent some of the most popular and long-standing methods, collectively with >3000 citations as of July 2022. Other categories of functional genomic annotation exist, such as methylation[22] and protein[23] QTLs, and were beyond the scope of the present analysis; we expect that their performance would not substantially differ from the methods evaluated here, but we cannot account for significant GWAS2 loci acting through other mechanisms whose functional annotations were not evaluated here. Other mechanisms may explain some of the low SN, but PPV would be unaffected.

Fourth, our definition of "gold standard" using GWAS2 hits assumes that all genome-wide significant variants in GWAS2 are truly trait-associated. In the modern GWAS era with independent replication as a best practice, this assumption likely holds for most loci and their variants. By evaluating variants and their broad flanking regions, our approach minimized FNs and FPs caused by changes in lead variants for a given locus across GWAS waves and equalized the playing field for variant- and eQTL/gene-based methods, allowing for simultaneous comparison.

Fifth, our definition of a FP depended on the sample size of GWAS2. Variants or loci nominated by functional weighting methods could be classified as FP when compared against the gold standard of GWAS2, but it is possible that they represent TP associations that GWAS2 remained underpowered to detect. By performing a sensitivity analysis using recently published GWAS3 as our gold standard for the three psychiatric traits, we confirmed that a portion of the genome classified as FP in our primary analysis, with GWAS2 as the gold standard, were trait-associated. In a real-world application, the ability to arrive at this conclusion would require either substantial laboratory follow-up or an increase in GWAS sample size of 2–4 times to bridge the gap between GWAS2 and GWAS3, using the psychiatric traits as representative sample sizes.

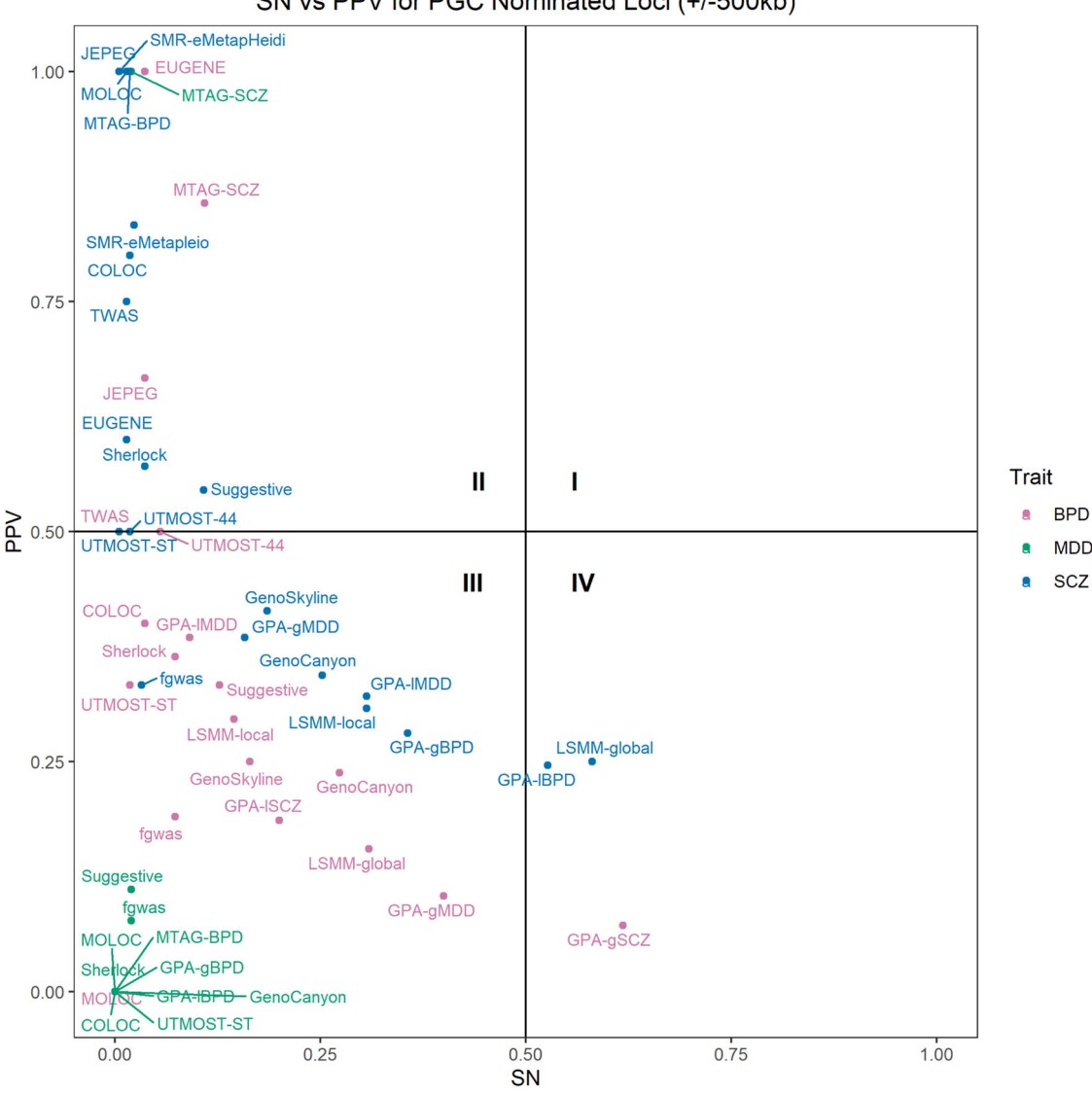

**Fig. 4 Relationship between sensitivity and positive predictive value for method-psychiatric trait combinations.** Scatterplot of the relationship between sensitivity (SN) and positive predictive value (PPV) for method-psychiatric trait combinations that return nominated variants. SN and PPV were calculated using $+/-500$ kb overlap criteria and compared to GWAS3 as the gold standard. Horizontal and vertical lines denote SN and PPV of 50%, respectively.

Functional annotation databases continue to expand and contribute broadly to understanding human biology and uncovering causal underpinnings of variant-trait associations. Although functionally weighting GWAS is not a substitute for pursuing large samples for well-powered GWAS, these summary statistics-focused methods can be a cost-efficient approach to discovery. Our results show that no method applied systematically across five traits produced both high SN and high PPV. Functional weighting GWAS methods might boost either SN or PPV where larger sample sizes are not feasible and the currently available GWAS has generated at least some genome-wide significant loci for the trait of interest. Greater tolerance for FPs can be endured by a research pipeline incorporating inexpensive, high-throughput, and/or in silico steps, while a pipeline intended to move GWAS nominations into model organisms may require more confidence that the nominated loci are truly trait-associated. Functional weighted GWAS results can generate leads for follow-up studies of the genetic drivers of complex traits with a reasonable likelihood of being true, particularly for associations that come through multiple methods.

## Methods
**Method selection**. We reviewed the published literature through February 2020 to identify methods that met the following criteria:

  i. Descriptively categorized as (a) annotation-based; (b) pleiotropy-based; or (c) eQTL-based.
 ii. Utilized GWAS summary statistics, as opposed to individual-level genotype data.
iii. Implemented using freely-available software or packages.
 iv. Provided either method-specific annotation or eQTL files for use with the method, or were amenable to use with publicly available annotation datasets (e.g., GTEx[46]).
  v. Originally proposed primary or secondary usage included the discovery of novel trait-associated variants, genes, eQTL, or loci.

We found 17 functional weighting methods that met our inclusion criteria. We also evaluated the performance of a "suggestive" $p$-value threshold, defined as $5 \times 10^{-8} \le p < 1 \times 10^{-5}$ to illustrate the tradeoffs of simply choosing a more liberal $p$-value cutoff, without the addition of any functional weighting

information. The full list of 18 methods evaluated in the present analysis is presented in Table 1. These methods varied in their determination of significant trait associations. For methods that listed specific threshold values for test statistics, we used those thresholds. For methods whose test statistics were $p$-values and whose authors did not provide a significance threshold, we used a Bonferroni correction on the number of valid $p$-values output by the method. Details for significant trait association determinations for each method are detailed below and in Supplementary Data 1.

*Suggestive*. For all traits, we considered "suggestive" variants as those with $p$-values $< 1 \times 10^{-5}$ and $\geq 5 \times 10^{-8}$. To define suggestive loci, we defined a region $+/-500$ kb surrounding the variant with the smallest suggestive $p$-value, and collapsed regions that overlapped by any amount into a single locus.

*GenoCanyon*. We downloaded the prediction scores for the human genome smoothed over 10-kb segments[47] (zhaocenter.org/GenoCanyon_Downloads.html) and applied them to each of the five GWAS1 using the signal prioritization software GenoWAP[48]. We used the recommended posterior probability of 0.50 to define statistical significance.

*GenoSkyline*. We downloaded tissue-specific functional predictions[49] (http://genocanyon.med.yale.edu/GenoSkyline) based on the Roadmap Epigenomics Project (Roadmap) for whole blood and brain tissue and applied them to the blood traits and psychiatric traits, respectively, using the signal prioritization software GenoWAP[48]. We used the recommended posterior probability of 0.50 to define statistical significance.

*Weighted eQTL*. Following the method of Li et al.[50], we calculated both binary and general eQTL-based weights for all five traits. In each case, we set $\alpha = 0.05$ and power $= 0.6$. For binary weights, the parameter M was the number of included variants in each GWAS1, respectively, and $\varepsilon$ was calculated as the percentage of eSNPs, defined as those with significant eQTL associations in the relevant tissue. We used the significant GTEx v7 Brain Nucleus Accumbens for PGC traits and the significant GTEx v7 Whole Blood for blood cell traits. Weights were then normalized and applied to the downloaded $p$-values. Statistical significance was defined as $p_{\text{weighted}} < 5 \times 10^{-8}$.

The general eQTL weight was calculated as $\sqrt{(-\log_{10} p_{\text{eQTL}})}$ for eSNPs and 1 for all others, where eSNPs are defined as above. Weights were then normalized and applied to the downloaded $p$-values. Statistical significance was defined as $p_{\text{weighted}} < 5 \times 10^{-8}$. The parameters $\alpha$, power, and M were also defined as above.

*GPA*. Genetic analysis incorporating Pleiotropy and Annotation[51] (GPA) was performed using pairwise comparisons between two traits of interest. For each pair of traits, we matched variants on hg19 chromosome and position. In the case of duplicate variants, the variant with the smaller $p$-value was retained. We performed GPA with both global and local FDR strategies using a cutoff of 0.05 in both cases to determine statistical significance.

For each of the three PGC traits, we used as pleiotropic traits the remaining two PGC traits. For blood cell traits, we used the second blood cell trait as a pleiotropic trait for the first. Additionally, we used SCZ1[52], BMI[53], height[54], and two GWAS for HDL[55,56] (http://www.nealelab.is/uk-biobank/). FDR cutoffs were defined as above in all cases.

*MTAG*. Multi-Trait Analysis of Genome-wide association summary statistics (MTAG[33]) was performed using pairwise comparisons between two traits of interest. For all traits, we used the subset of downloaded variants with valid rsIDs and allele frequencies in the downloaded GWAS1 summary statistics. Statistical significance was defined as $p_{\text{MTAG}} < 5 \times 10^{-8}$. For each of the three PGC traits, we used as pleiotropic traits the remaining two PGC traits. For blood cell traits, we used the second blood cell trait as a pleiotropic trait for the first. Additionally, we used SCZ1[52], BMI[53], height[54], and two GWAS for HDL[56,57] (http://www.nealelab.is/uk-biobank/).

*fgwas*. We combined fgwas[31] with eQTL results from GTEx[46] as the annotation database. Our eQTL dataset of choice was the significant eQTL dataset for expression in the Nucleus Accumbens from GTEx, v7 for PGC traits and the significant GTEx v7 Whole Blood eQTL dataset for blood cell traits. Each significant eQTL was defined as a "segment". All GWAS1 variants whose position fell within the start and end positions[58] of a significant eQTL were assigned to that segment. Variants that remained unassigned to any segment were excluded, along with variants having missing allele frequencies or odds ratios of zero in the downloaded summary statistics. If more than one variant was localized to the same position in GWAS1, the variant with the smallest $p$-value was retained. We used the default likelihood penalty of 0.2 to run fgwas. Statistical significance was defined as a PPA $> 0.9$.

*Sveinbjornsson*. We applied the functional-weighted GWAS method described by Sveinbjornsson et al.[59], dubbed here the "Sveinbjornsson" method. This method relies on an annotation classification for each variant into one of four categories, where each category has a Bonferroni-adjusted family-wise error weight reflecting the likelihood of protein function alterations caused by that variant. The categories and $p$-value thresholds are loss-of-function ($p < 5.5 \times 10^{-7}$), moderate impact ($p < 1.1 \times 10^{-7}$), low impact ($p < 1.0 \times 10^{-8}$), and other ($p < 1.7 \times 10^{-9}$). We annotated GWAS1 summary statistics using SnpEff software[60] and applied the aforementioned $p$-value cutoffs according to the annotation category to determine statistical significance.

*LSMM*. We performed latent sparse mixed model (LSMM[61]) following the example annotations of the method authors, requiring three sets of input: variants and $p$-values from GWAS summary statistics, ANNOVAR[62], and GenoSkylinePlus[63] using annotations from the original source. We downloaded the hg19 annotations from the ANNOVAR website and used the dbSNP147 database to annotate GWAS1 variants. Annotations were then collapsed into nine categories: downstream, exonic, intergenic, intronic, ncRNA/exonic, ncRNA/intronic, upstream, 3'UTR, and 5'UTR, with each variant assigned a value of 0 or 1 to denote category membership.

*COLOC*. For PGC traits, our colocalization[25] dataset of choice was the significant eQTL dataset for expression in the Nucleus Accumbens from GTEx, v7[46]. We defined a region to test for colocalized signal as $+/-200$ kb upstream and downstream from start and stop positions of a single eQTL probe, and included all GWAS1 SNPs contained within that region. This was repeated for all eQTL probes available in the downloaded dataset.

For blood cell traits, we repeated the same procedure using the significant Whole Blood GTEx v7 dataset[46]. For all GWAS1, evidence of statistically significant colocalization was defined as an Approximate Bayes Factor greater than 0.75.

*ENLOC*. We performed the fastENLOC implementation of the ENLOC method[64]. We downloaded the multi-tissue eQTL annotation derived from GTEx v8[65] hg38 position and provided European LD definition file (https://github.com/xqwen/fastenloc/). We

then used LiftOver[66] to convert all five GWAS1 from hg19 to hg38 genomic coordinates. We applied the Nucleus Accumbens eQTL dataset For PGC traits and the Whole Blood eQTL dataset for blood cell traits.

*EUGENE.* For all traits, we used the subset of downloaded variants with valid rsIDs as GWAS1. We downloaded the required input datasets for gene position from the EUGENE website[67], grouped GTEx brain tissues as the eQTL data for the PGC traits[46], and grouped whole blood eQTL data for the blood cell traits[46,68,69] after performing additional quality control on the whole blood eQTL data to remove discrepant rsIDs. We used Satterthwaite's approximation to calculate the gene-based summary statistics[70]. We then estimated the FDR thresholds using EUGENE and identified the *p*-value threshold closest to the FDR threshold of 0.05 to determine statistical significance (Supplementary Data 1).

*JEPEG.* We downloaded the SNP annotation data (v0.2.0) and reference panel (1000 Genomes EUR Phase 1 Release 3)[71] from the JEPEG website (https://dleelab.github.io/jepeg/). For all traits, we used the subset of downloaded variants with valid rsIDs. For blood cell traits, in the case of duplicate rsIDs, we retained the variant with the smaller *p*-value. Statistical significance of the results was determined by a Bonferroni correction applied to the JEPEG *p*-value.

*MOLOC*[25]. For PGC traits, our colocalization dataset of choice was the significant eQTL dataset for expression in the Nucleus Accumbens from GTEx, v7[46]. The methylation dataset used for PGC traits was downloaded from the processed data available on the GEO data repository at accession number GSE74193 and reflects the identification of meQTLs in the prefrontal cortex of 191 schizophrenia patients and 335 controls without psychiatric illness[72]. We defined a region to test for colocalized signal as +/−200 kb upstream and downstream from start and stop positions of a single eQTL probe, and included all GWAS1 SNPs and meQTL probes contained within that region. This was repeated for all eQTL probes available in the downloaded dataset.

For blood cell traits, we repeated the same procedure using the significant Whole Blood GTEx v7 dataset. The methylation dataset used for blood cell traits was the methylation QTL results from the ALSPAC Accessible Resource for Integrated Epigenomics Studies (ARIES)[73] at the middle-aged timepoint (https://data.bris.ac.uk/data/dataset/r9bxayo5mmk510dczq6golkmb).

*Sherlock.* Only variants with valid rsIDs were submitted to Sherlock for each GWAS. For the PGC traits, the Sherlock-provided eQTL data was chosen as GTEx v7 Brain – Nucleus accumbens. Sample sizes from Supplementary Data 2 were used, and disease prevalence was taken as 0.5% for schizophrenia[74], 1% for bipolar disorder[75], and 15% for major depressive disorder[76]. For the UK Biobank traits, the Sherlock-provided eQTL data was chosen as GTEx v7 Whole Blood, and the sample sizes from Supplementary Data 2 were entered for sample size. As Sherlock output is sometimes presented as a gene symbol and sometimes as an Ensembl gene ID (ENSG), we used the GENCODE annotations[58] to match gene symbols to Ensembl IDs and evaluated the overlap with our gold standards using Ensembl IDs.

*SMR.* For all GWAS1 inputs, we used the subset of downloaded variants with valid rsIDs and valid allele frequencies. Formatted eQTL data were downloaded from the SMR website (https://cnsgenomics.com/software/smr/#DataResource). For PGC traits, we evaluated the performance of SMR using three different eQTL datasets: GTEx v7 data from the Brain Nucleus Accumbens[46], the

"lite" version of the GTEx v7 data from the Brain Nucleus Accumbens, and the Brain-eMeta eQTL data[77] derived from a meta-analysis of GTEx brain, Common Mind Consortium[78], and ROSMAP consortium[79] studies. For UK Biobank traits, we evaluated the performance of SMR using the GTEx v7 data from Whole Blood[46], both full and "lite" versions. For all datasets, we evaluated with and without the requirement for a Heidi *p*-value of <0.05 to exclude trait-eQTL associations due to pleiotropy.

*TWAS.* For all GWAS1 inputs, we used the subset of downloaded variants with valid rsIDs. We downloaded reference LD data for the 1000 Genomes EUR samples provided by the Broad Institute Alkes Group (https://data.broadinstitute.org/alkesgroup/FUSION/). We downloaded (https://gusevlab.org/projects/fusion/) and applied pre-computed gene expression weights for GTEx v7 Brain Nucleus Accumbens for PGC traits and Whole Blood for blood traits[46].

*UTMOST.* For all GWAS1 inputs, we used the subset of downloaded variants with valid rsIDs. We used the pre-calculated covariance matrices using the 44 GTEx v7[46] tissues (https://github.com/Joker-Jerome/UTMOST). For all five GWAS1, we evaluated the full cross-tissue expression UTMOST results. We additionally evaluated the single-tissue UTMOST output for the Nucleus Accumbens for PGC traits and Whole Blood for blood traits.

**Model trait selection**. We evaluated the performance of the functional weighting methods using five traits meeting the following criteria:

1. At least two "waves" of GWAS summary statistics that were publicly available for download.
2. Both available waves were conducted in the same ancestral population.
3. The waves had to differ in sample size to such a degree that the larger GWAS contained more genome-wide significant associations than the smaller wave.

We deliberately chose early phase GWAS (which we refer to as GWAS1) for each trait to allow for validation of results in subsequent GWAS for the traits (referred to as GWAS2 and/or GWAS3). We evaluated three traits with summary statistics available from the Psychiatric Genomics Consortium (PGC): schizophrenia[52] (SCZ), bipolar disorder[80] (BPD), and major depressive disorder[14] (MDD). We also evaluated two blood cell traits examined in the UK Biobank, mean platelet volume (MPV) and white blood cell count (WBC)[81], as examples of traits with a larger explained heritability, many genome-wide significant loci, minimal heterogeneity in phenotyping, and comprehensive tissue-specific functional annotations. In addition to meeting the above criteria, these five traits, collectively, represent the spectrum of GWAS discovery, from no statistically significant variants to identifying nearly all common variation in the studied population. Additional details of the GWAS used to test the functional weighting methods are presented in Supplementary Data 2.

We used Liftover[82] to convert the GWAS of the psychiatric traits from hg18 to hg19. For MPV, *p*-values were truncated at $7.41 \times 10^{-323}$, due to the extremely small *p*-values not being read into R (v3.6.0).

**Definition of a gold standard**. For comparison to each GWAS1, we used as our "gold standard" a larger, more powerful GWAS, hereafter referred to collectively as GWAS2, performed on the same trait and by the same consortium to reduce variability in

findings due to differences in trait definition, analytic strategies, or recruitment of study participants (Supplementary Data 2). Our gold standard "hits" for each GWAS2 were defined as those variants meeting the standard genome-wide significance threshold of $5 \times 10^{-8}$. We defined a significant locus as the region extending $+/-500$ kb from the variant with the smallest $p$-value. Additional variants with genome-wide significant $p$-values within this region were included within the locus of the lead variant. This procedure was repeated in a stepwise fashion until all genome-wide significant variants were captured. As a final step, overlapping one-megabase intervals were combined into a single locus, and the extended HLA region was defined as the region spanning from base pair 25,000,000 to 35,000,000 on chromosome 6.

**Exclusion of significant GWAS1 hits**. All GWAS1 contained statistically significant loci except for MDD (Supplementary Data 2). To avoid giving credit to the functional weighting methods for "re-discovering" these significant loci, we excluded them from evaluation after applying the functional weighting method to GWAS1. We defined a significant locus in GWAS1 as the region extending $+/-500$ kb from the variant with the smallest $p$-value. Additional variants with genome-wide significant $p$-values within this region were included within the locus of the variant of the smallest $p$-value. This procedure was repeated in a stepwise fashion until all genome-wide significant variants were accounted for. As a final step, overlapping one-megabase intervals were combined into a single locus and the extended HLA region was defined as above.

To exclude GWAS1 hits from the set of GWAS2 "gold standard" hits available for discovery, we used the GenomicRanges R package[83] to remove from GWAS2 any loci with any degree of overlap with the defined GWAS1 significant loci.

**Evaluation metrics**. Because we focused on method performance to discover novel GWAS hits, our evaluations were based on calculating sensitivity (SN), positive predictive value (PPV), and the F1 score (F1, the harmonic mean of SN and PPV). Definitions can be found in Supplementary Data 3. Because variants with non-significant $p$-values in GWAS1 may be truly associated with the trait, but GWAS1 was not statistically powerful enough to uncover their associations, we avoided evaluation metrics that depend on the definition of a TN.

**Evaluation of variant-level methods**. Nine functional weighting methods, including the use of a suggestive $p$-value threshold, provided results for individual genetic variants (Table 1). To evaluate the performance of these nine methods on a per-variant level, TP variants were defined as those with matching chromosome and position that were both genome-wide significant in GWAS2 and nominated as significant by the functional weighting method either by the threshold specified by the method or, if no threshold was explicitly stated, by a Bonferroni multiple testing-corrected threshold (Supplementary Data 1). To exclude variants that were statistically significant in GWAS1, we excluded variants within the $+/-500$ kb boundaries of GWAS1 hits defined above (see the section "Exclusion of significant GWAS1 hits").

Because the functional weighting methods cannot account for secondary signals and some do not account for linkage disequilibrium, we also calculated SN, PPV, and F1 using a locus-based definition of statistical significance. In this evaluation, we defined each locus in the same manner as we identified GWAS1 significant hits (see the section "Exclusion of significant GWAS1 hits"). A TP was defined as an overlap of at least 250 kb

in the 1 MB flanking window of a top locus in GWAS2, as defined above, and a $+/-500$ kb window of a variant nominated by a functional weighting method. A FP was defined as a $+/-500$ kb window nominated by a functional weighting method with less than 250 kb overlap among any GWAS2 loci. A FN was defined as a GWAS2 locus with less than 250 kb overlap with any locus nominated by the functional weighting method being evaluated. This locus-to-locus comparison was performed assessing any degree of overlap between the gold standard GWAS2 loci, excluding the significant GWAS1 loci, and the loci calculated from the results of the functionally weighted GWAS1 using the GenomicRanges package[83] in R.

**Evaluation of eQTL-level methods**. To comparably evaluate methods that yield results on the level of eQTL or gene, we calculated transcript- or gene-based $p$-values using MAGMA[84]. As most of the eQTL data used in these comparisons came from GTEx, we downloaded and used their GENCODE annotations[58] for transcript/gene names and genomic locations. Statistical significance for GWAS2 was determined at a GWAS-specific Bonferroni correction to the MAGMA $p$-value after excluding eQTL-based gene results that did not yield a MAGMA $p$-value.

For methods that did not provide a significance threshold, we first excluded any results that did not result in a valid statistic, then performed a Bonferroni correction based on the number of remaining tests. To exclude established significant loci from GWAS1, we excluded nominated transcripts/genes where the midpoint of the genomic location was within $+/-500$ kb of the GWAS1 loci, defined above (see the section "Exclusion of significant GWAS1 hits"). We did not exclud genes from either GWAS1 or GWAS2 MAGMA results.

For MAGMA-based evaluations, TP, FP, and FN were determined by matching either the Ensembl ID or gene symbol, depending on what was used by the particular functional weighting method (Supplementary Data 1), to the output of our modified MAGMA analysis to each GWAS2. A TP was defined as an eQTL/gene that was nominated as significant by the functional weighting method and identified as statistically significant by MAGMA as described above. FP and FN were defined analogously, and we calculated SN, PPV, and F1.

We also conducted locus-based evaluations in two other ways. The first was to use the boundaries of the nominated eQTL/gene, either defined by the functional weighting method when provided or the GENCODE annotation boundaries used to generate the MAGMA $p$-values (Supplementary Data 1). The second approach was to define the locus boundaries for a functionally weighted eQTL/gene as $+/-500$ kb from the midpoint of the previously stated boundaries. To avoid possible double-counting, we merged overlapping eQTL/genes into a single locus. Loci were determined in a similar fashion as before (see the section "Exclusion of significant GWAS1 hits") using the midpoint of the GENCODE-defined start and end positions, with no truncation at the ends of chromosomes or centromeres, with the exception of EUGENE, where we used the chromosome and position defined by EUGENE output.

We performed a locus-to-locus comparison by looking for a minimum of 250 kb of overlap when nominations were defined as $+/-500$ kb from the midpoint, and 2500 bases of overlap when nominations were defined by the start and end positions using the GENCODE annotation boundaries between the gold standard loci calculated from GWAS2 (see the section "Definition of a gold standard") and the loci calculated from the results of the functionally weighted GWAS1 using the GenomicRanges package[83] in R.

**Generation of UpSet plots.** To identify an optimal ensemble approach, we examined the overlap among nominations across functional weighting methods for each trait by generating UpSet plots. Plots were generated using the ComplexUpset package[85,86] in R. To construct the UpSet plot, for each trait, functional weighting GWAS methods were ordered from largest to smallest number of nominated loci, defined using +/−500 kb from either the top variant or gene midpoint. For methods with multiple options, the top performing option was selected based on largest PPV. A matrix of nominated loci vs the fwGWAS methods was created in a stepwise fashion. The method nominating the largest number of loci was populated first, and then each of its nominated loci was tested for overlap of at least 250 kb with loci nominated by all other methods and these overlaps populated the matrix. For each subsequent functional weighting GWAS method, only nominated loci that had not been found to overlap with loci from previously examined methods were added to the matrix. These new additions were then checked for overlap with all remaining method nominations, and all methods nominating a new locus were noted on the matrix.

**Application of 17 functional weighting methods and a suggestive threshold to model traits.** Full details of the application of each functional weighting method can be found in the Supplementary Note 1, with details of significance cutoffs and functional databases presented in Supplementary Data 1. Briefly, we used the default inputs, external databases, and statistical significance cutoffs recommended by the method developers to the full extent that they were provided. When statistical significance cutoffs were not provided, we applied a standard threshold of either a Bonferroni-corrected $p$-value or a false discovery rate cutoff of 0.05, as appropriate for the statistics calculated by the functional weighting method.

For the choice of functional database to use with each method, our default was to use a preformatted database provided by the method developers (e.g., TWAS/FUSION). When multiple databases were made available (e.g., SMR), we chose the largest database representing a tissue type appropriate to the model trait being evaluated.

When no functional database was made available by the method authors (e.g., COLOC), we used the statistically significant GTEx v7 nucleus accumbens data downloaded from the GTEx data portal to apply the functional weighting methods to the three psychiatric traits[87–89] and the corresponding statistically significant GTEx v7 whole blood data for the two blood cell traits[46].

We investigated the performance of the pleiotropy-based methods GPA and MTAG in each psychiatric trait using contemporaneous GWAS of the other two psychiatric traits. For the blood cell traits, we used a variety of potentially omnigenetic traits: SCZ[52], HDL cholesterol[56,57] (http://www.nealelab.is/uk-biobank/), BMI[53], height[54], and the other blood cell trait[81].

**Sensitivity analyses.** To determine whether the 250 kb overlap between a nomination and a novel GWAS2 locus impacted our results, we tested overlaps of 1 base, 500 kb, and 750 kb, used to replace the +/−500 kb and Ensembl locus definitions.

We investigated whether a less stringent cutoff resulted in better performance by applying an FDR significance cutoff for those methods (suggestive, MTAG, Weighted eQTL, JEPEG, TWAS/FUSION, and UTMOST) for which we used a Bonferroni multiple testing correction. The FDR correction was implemented using the fdrtool R package[90] and a cutoff of $q < 0.05$ was used to determine statistical significance.

We sought to determine the accuracy of our FP definition by using wave 3 GWAS, hereafter referred to as GWAS3, recently released by the PGC for the three psychiatric traits. Known loci from GWAS1 were excluded as described above.

**Reporting summary.** Further information on research design is available in the Nature Portfolio Reporting Summary linked to this article.

## Data availability
All data used in this study has been previously published and all methods are freely available. A complete list of links to data and methods used in this study is available in Supplementary Data 9. Data used to compile scatterplots and UpSet plots are available as Supplementary Data 10 and 11, respectively.

## Code availability
Code used to generate scatterplots and UpSet plots are provided as Supplementary Notes 2 and 3, respectively. Additional code available upon request.

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

## Acknowledgements

This work was funded by NIDA R01 DA044014: Harnessing Knowledge of Gene Function in Brain Tissue for Discovering Biology Underlying Heroin Addiction.

## Author contributions

A.M. analyzed data and prepared the manuscript. J.A.M., B.C.Q., and Y.G. assisted with data analysis. E.O.J., D.B.H., G.P.P., and N.C.G. conceived of the project, oversaw the project, and revised the manuscript. L.J.B. revised the manuscript. All authors read and approved the final manuscript.

## Competing interests
The authors declare no competing interests.
