## [Peer Review File · Communications Biology]

Reviewers' comments:

Reviewer #1 (Remarks to the Author):

This is an interesting paper that systematically evaluates 18 published functional weighting methods to facilitate the identification of genetic signals from GWAS studies. The idea being tested is that using these approaches, potentially underpowered GWAS may be able to provide additional insight beyond standard GWAS analyses approaches which may be useful were additional case recruitment is difficult. Five traits were examined in the evaluations, schizophrenia, bipolar disorder, MDD, mean platelet volume and white blood cell count. Overall, the findings show that no method achieved both high sensitivity and positive predictive value and the methods cannot overcome the need for large well designed GWAS, however there were some novel loci identified.

While a useful study and timely as GWAS data becomes more available, there is a need for several aspects to be addressed prior to potential publication.

Main comments

1. The choice of traits as test cases is unclear and needs some clarity, justification.
2. Why only 5 traits? It would be useful to see how performs more broadly.
3. The use of a +/-500bp window needs some comment, justifications, discussion. Where other windows assessed and what was the outcome?
4. It is unclear what the novel findings were (e.g. identified by multiple methods) and so the paper should include a table/section devoted to potential new findings related to the traits studied.
5. Full outputs of the novel variants identified should be included.

Specific comments requested:

1. Extensive information on methods, including statistics is provided.
2. The paper is useful and may influence the field in the consideration of these methods.

Reviewer #2 (Remarks to the Author):

The GWAS analysis provided valuable genetic information of phenotypes. And there were many methods in the subsequent analysis, including annotation, gene-base analysis, gene-set analysis, eQTL and so on. And the study reviewed current methods about improving GWAS statistical power to uncover novel loci, which was very interesting. I do however have some comments for this manuscript:

1. As the introduction mentioned, the purpose of the article was to identify suitable method(s) to improve GWAS statistical power to uncover novel loci. The article reviewed these methods and divided into three classification including annotation, pleiotropy and eQTL. Is it too complicated to summarize all 18 methods together due to different purpose between annotation, pleiotropy and eQTL? Please give relevant reasons about these.
2. The article provided some references about the brief introduction of 18 methods. The purpose was to identify suitable method(s) for improving GWAS statistical power to uncover novel loci. However, there were not more information about how these 18 methods could improve power, please provide relevant supplementary materials.
3. The article applied PPV and FN to assess the performance of 18 methods. To our knowledge, there were more approaches to evaluate the performance, such as AUC (Chung et al, doi: 10.1371/journal.pgen.1004787), NRI (Net reclassification index) and so on. Why not try to use similar methods?

Minor question:

1. The question about method Selection (p7:272-278), please use different numeric ordinal levels to better distinguish the subtitle levels, such as 1) Categorized as a) annotation-based; b) pleiotropy-based; or c) eQTL-based; 2) Utilized GWAS summary statistics, as opposed to individual-level genotype data and so on.
2. Please check carefully the order of tables provided, such as p11: 468-469 (Additional details of the GWAS used to test the functional weighting methods are presented in Supplemental Table 1). The detailed information of GWAS used to test the functional weighting methods was provided in Supplemental Table 2.

Reviewer #3 (Remarks to the Author):

The authors present a comprehensive evaluation of functional weighting methods for identifying novel associations given GWAS summary statistics. The ongoing interest in GWAS and related down-stream analyses results in a multitude of potential applications of the study's outcome. For this reason this manuscript is certainly of interest for the readership of COMMSBIO.

MAJOR COMMENTS:

- Given the literature review has been conducted in 02/2020, I suggest to extend Table 1 by incorporating more recent methods (e.g. PLEIO [Lee et al. (2021)]). Furthermore, I recommend to update citation count and/or date of retrieval for the methods under investigation (>3,000 citations as of 10/2021).
- Selecting appropriate data for evaluating the weighting methods seems crucial. Therefore I suggest to include (an extended version of) Supplementary Table 2 into the main manuscript. Furthermore, I recommend to discuss the following questions: (a) if sample size of two wave 1 studies differs considerably, how does this influence the outcome of the method evaluation? (b) if increase of sample size from wave 1 to wave 2 differs considerably between two diseases, how does this influence the outcome of the method evaluation? (c) if number of genotyped markers of two studies from wave 1 or wave 2 differs considerably, how does this influence the outcome of the method evaluation? (d) if increase of number of genotyped markers from wave 1 to wave 2 differs considerably between two diseases, how does this influence the outcome of the method evaluation?
- As shown in Supplementary Table 4, for a given GWAS the number of input markers varies considerably between the different weighting methods (e.g. BPD: 2422487 (GWAS1 suggestive) vs. 188570 (fgwas), SCZ: 1242114 (GWAS1 suggestive) vs. 885859 (MTAG Using BPD)). How do these differences affect the outcome of the method evaluation? How would appropriate subsampling of markers affect the result?
- I recommend to order the methods listed in Table 2 in accordance to their median rank. Furthermore, I suggest to extend Table 2 by variability estimates of PPV, e.g. generated based on the above-mentioned subsampling of markers.

MINOR COMMENTS:

- "[...] when focused on either high SN or high PPV, functional weighting GWAS methods boost statistical power [...]". This sentence partially contradicts with statements about methods with performances in Quadrant II of Figure 1 and 4 (low SN and high PPV), for which no boost in statistical power (=SN) has been observed. The following statement might be more appropriate: "Functional weighting GWAS methods might boost either SN or PPV where larger sample sizes are not feasible and the currently available GWAS has generated at least some genome-wide significant loci for the trait of interest."
- Please ensure concordance to the COMMSBIO Style and formatting guide by (a) renaming Citations to References, (b) adding a statement on Author contributions, (c) adding a statement on Competing

interests, (d) labelling and citing Supplementary items as Supplementary Figure 1, Supplementary Table 1, Furthermore, please check whether "Neale, B. M. No Title. <http://www.nealelab.is/uk-biobank/>." corresponds to the Nature referencing style.

- I recommend to check Supplementary Table 1, 2, 6, 7 for correct cell contents (e.g. "ENSG locus calculated by...", "500kb locus calculated by...", "de Leeuw, et al. PLoS Comput Biol. 2016;11(40:e1004219)", consistent formatting (e.g. borders, alignment), explanations for abbreviations (e.g. Supplementary Table 1: SCZ, BPD, MDD, MPV, WBC, HDL, PGC, UKBB, CHR, POS, BETA, SE, N, MAF, ProbeID, A1, A2, A1AF, P), explanations for highlighting (e.g. Supplementary Table 6: highest ranks marked in bold font) and removal of unrelated information (e.g. Supplementary Table 2, B55:C60).

Reviewer 1	Responses
(Remarks to the Author): This is an interesting paper that systematically evaluates 18 published functional weighting methods to facilitate the identification of genetic signals from GWAS studies. The idea being tested is that using these approaches, potentially underpowered GWAS may be able to provide additional insight beyond standard GWAS analyses approaches which may be useful were additional case recruitment is difficult. Five traits were examined in the evaluations, schizophrenia, bipolar disorder, MDD, mean platelet volume and white blood cell count. Overall, the findings show that no method achieved both high sensitivity and positive predictive value and the methods cannot overcome the need for large well designed GWAS, however there were some novel loci identified. While a useful study and timely as GWAS data becomes more available, there is a need for several aspects to be addressed prior to potential publication.	
1. The choice of traits as test cases is unclear and needs some clarity, justification.	We chose traits that allowed us to represent the spectrum of GWAS discovery, from no statistically significant variants (MDD GWAS1) to nearly all common variation identified (MPV GWAS1). In order to perform the evaluation of these methods, the traits also had to meet several additional criteria, including 1) at least two “waves” of GWAS summary statistics were publicly available, 2) these two waves had to be in the same ancestral population, and 3) had to differ in sample size to sufficient degree that the larger GWAS contained more genome-wide significant associations than the smaller wave. We added these criteria to the Methods section, lines 485-490. Previously, it read “We evaluated the performance of the functional weighting methods using five traits that have published GWAS with publicly available summary statistics.” Revised, it reads “We evaluated the performance of the functional weighting methods using five traits meeting the following criteria:

	 1. At least two “waves” of GWAS summary statistics that were publicly available for download 2. Both available waves were conducted in the same ancestral population 3. The waves had to differ in sample size to such a degree that the larger GWAS contained more genome-wide significant associations than the smaller wave.”
2. Why only 5 traits? It would be useful to see how performs more broadly.	Our goal was not to generate a database of reweighted association statistics for all publicly available GWAS, as some have done (LDhub, neale lab, etc). Furthermore, as some of these methods are quite popular for in silico follow-up of GWAS hits, to do so would not necessarily report novel findings. Instead, we chose traits that allowed us to represent the spectrum of GWAS discovery, from no statistically significant variants (MDD GWAS1) to nearly all common variation identified (MPV GWAS1). We assert that these five traits are sufficiently representative of highly polygenic traits and that our findings can be generalized broadly. We have added the following text to the Methods section, lines 498-501: “In addition to meeting the above criteria, these five traits, collectively, represent the spectrum of GWAS discovery, from no statistically significant variants to identifying nearly all common variation in the studied population.”
3. The use of a +/-500bp window needs some comment, justifications, discussion. Where other windows assessed and what was the outcome?	The use of +/- 500 bases as an interval to define independent loci was based on the use of this interval size by several GIANT consortium (Locke et al., 2015; Shungin et al., 2015). Although other rational intervals can be proposed, we did not evaluate the interval choice as a source of variability. Instead, we considered several definitions of “true positive” depending on whether the functional weighting method generated results at the variant level or at the eGene level. We also performed a sensitivity analysis examining the effect of required overlap size necessary to call a nominated locus a “true positive”. Defining a locus as +/- 500 kilobases from a particular point was our answer to the question

	of how to harmonize the definition of a “true positive” across variant-level and eGene-level methods; we found this to be a reasonable solution to a real problem in this area of method evaluation. We are not aware of any other publication that has compared both variant-level and eGene-level methods in a quantitative fashion. The work necessary to find an optimum locus definition that unifies variant-level and eGene-level methods is beyond the scope of this analysis. We added the following section to the results, lines 137-147: “Effect of Evaluation Strategy Only one evaluation strategy allowed for direct comparison between variant-level and eQTL-level methods; however, we also considered several strategies that permitted us to rank variant-level and eQTL-level methods amongst themselves, respectively. Among variant-based method evaluations, the top method for SN changes from LSMM to a tie between LSMM and GPA when evaluated on a variant-to-variant comparison; GPA was ranked second by the 500kb comparison. There were no changes in the top performing method for PPV. Among the eQTL-based method evaluations, there are no changes across the three ways to calculate SN (Supplemental Table 5). For PPV, changing the locus definition from the 500kb to the eQTL locus changes a 4-way tie between COLOC, UTMOST, Sherlock, and MOLOC to a three-way tie of the same excluding UTMOST; however, using Magma to evaluate performance, the top performer is JEPEG, which is otherwise in the middle of the pack for PPV. “
4. It is unclear what the novel findings were (e.g. identified by multiple methods) and so the paper should include a table/section devoted to potential new findings related to the traits studied.	As shown in our response to Reviewer 2, Comment 1, a primary motivating factor for our study was that many authors made claims that applying their methods to GWAS results identified novel trait-associated variants or genes, but did not offer independent confirmation that their “novel” findings were indeed trait-associated vs spurious. We sought to confirm these statements by other authors and provide comparative assessment of such methods.

	In the context of our analysis, a “novel” finding is defined relative to GWAS1. For our purposes, a novel trait-associated variant, gene, or locus had to be confirmed as genome-wide significant in GWAS2. Thus, our “novel” associations are not truly novel to the scientific community, rather, they represent the counterfactual scenario where an association that was not identified until the publication of GWAS2 could have been identified earlier in time by applying a functional weighting method to GWAS1. If a variant, gene, or locus is nominated by the application of a functional weighting method to GWAS1 but is not confirmed as trait associated in GWAS2, then we consider that a potential false positive rather than a “novel” finding. We discuss the implications of this assumption performed a sensitivity analysis using GWAS3, discussed in the Results section “Evaluating False Positives”, lines 148-161; and in the Discussion section lines 251-259. To address this comment and reduce reader confusion, we have reduced the number of times we use the word “novel” in our paper by replacing it with more descriptive language throughout when it previously referred to loci that were genome-wide significant in GWAS2 but not in GWAS1. We retain the word “novel” when referring to the general goal of our study, which is indeed to increase novel trait-association discoveries by evaluating methods that purport to do just that.
5. Full outputs of the novel variants identified should be included.	As stated in response to the previous comment, in the context of our analysis, a “novel” finding is defined relative to GWAS1. We are not asserting that any variants, genes, or loci nominated by applying functional weighting methods to GWAS1, be they defined as True Positives or False Positives according to our definitions outline in Supplemental Table 3, should be treated as definitive associations. In the case of True Positives, they are confirmed by GWAS2 and therefore not novel to the scientific community. In the case of False Positives, we do not claim that these variants, genes, and loci meet the conventional standards for

	identification as trait-associated. As demonstrated by our sensitivity analysis using GWAS3 as our gold standard, some fraction of these False Positives are indeed trait-associated, but the status of the remaining is uncertain.
Specific comments requested: 1. Extensive information on methods, including statistics is provided. 2. The paper is useful and may influence the field in the consideration of these methods.	

Reviewer #2	Responses
(Remarks to the Author): The GWAS analysis provided valuable genetic information of phenotypes. And there were many methods in the subsequent analysis, including annotation, gene-base analysis, gene-set analysis, eQTL and so on. And the study reviewed current methods about improving GWAS statistical power to uncover novel loci, which was very interesting. I do however have some comments for this manuscript:	
1. As the introduction mentioned, the purpose of the article was to identify suitable method(s) to improve GWAS statistical power to uncover novel loci. The article reviewed these methods and divided into three classification including annotation, pleiotropy and eQTL. Is it too complicated to summarize all 18 methods together due to different purpose between annotation, pleiotropy and eQTL? Please give relevant reasons about these.	We distinguish between annotation, pleiotropy, and eQTL methods primarily for descriptive purposes. The methods are distinguished by these categories when described in Table 1 and Supplemental Table 1. All three categories of methods' results are presented together without distinction in Figure 1, Figure 4, Supplemental Figures 1-4, and Supplemental Tables 4-8. The primary distinction in Supplemental Tables 4 and 5 is variant-level versus eGene-level evaluations. In the discussion, we changed line 256. Previously, it read "Third, our study focused on comparing three categories of methods: annotation, pleiotropy, and eQTL." It now reads "Third, our study focused on comparing three categories of methods which we describe as annotation, pleiotropy, and eQTL." We also changed Methods line 299. Previously, it read, "1. Categorized as a) annotation-based; b) pleiotropy-based; or c) eQTL-based". It now reads "1. Descriptively categorized as a) annotation-based; b) pleiotropy-based; or c) eQTL-based".

2. The article provided some references about the brief introduction of 18 methods. The purpose was to identify suitable method(s) for improving GWAS statistical power to uncover novel loci. However, there were not more information about how these 18 methods could improve power, please provide relevant supplementary materials.	We have added to Table 1 example text taken from a publication associated with each method demonstrating that the method authors considered their method to be useful to increase the number of trait-associated variants, genes, and/or loci. In recognition of the fact that not all method authors specifically mention statistical power, indicating the probability of correctly rejecting a null hypothesis, though we do use the phrases “statistical power” and “power” to indicate such, we have rewritten the following in our manuscript: Lines 54-56: Previously “Some authors suggest that a secondary usage of these methods is to increase the statistical power of GWAS to identify novel loci.” To “Some authors suggest that a secondary usage of these methods is to augment the ability of a given GWAS to identify novel trait-associated loci.”
3. The article applied PPV and FN to assess the performance of 18 methods. To our knowledge, there were more approaches to evaluate the performance, such as AUC (Chung et al, doi: 10.1371/journal.pgen.1004787), NRI (Net reclassification index) and so on. Why not try to use similar methods?	We believe there is a typo in the reviewer’s comment and that “FN” should be “SN”, or sensitivity, and we have responded as such. Our focus on the SN and PPV reflects our interest in two very specific, real-world questions of the methods study here. Of the nominated loci, what fraction are truly trait-associated (PPV)? Of the set of all trait-associated loci, what fraction are identified by a given method (SN)? These metrics provide relatable guidelines to future researchers who are likely to find themselves seeking additional trait-associated loci in one of two contexts. In the first, a researcher may seek to perform in vivo functional follow-up using expensive resources, such as a CRISPR-Cas9 mouse model, and may prioritize high PPV as the best way to increase the chances of meaningful results. In the second scenario, a researcher may instead seek to perform additional in silico or in vitro follow-up that is both high-throughput and inexpensive, and can therefore afford to tolerate the inclusion of more false positives but would benefit from a larger set of putative trait-associated loci, and prioritize methods that yield high SN. There are indeed evaluation metrics that we did not focus on, most of which can be derived from a basic confusion matrix. In choosing the

	metrics that were most appropriate for the present study, we recognized that the GWAS represents both an imbalanced classification problem and one in which any definition of a “negative”, that is, the set of variants/genes/loci that are not trait-associated, in reality contains a combination of truly non-associated loci and loci that are truly trait-associated but have not yet met the threshold for genome-wide significance with the sample size at hand. Therefore, we decided not to focus on evaluation metrics that describe the probability of correctly identify “true negatives”. This eliminates specificity and negative predictive value as metrics of interest and therefore implies that we should not calculate the AUC under an ROC or a PRC curve. Additionally, calculating an AUC under an ROC curve in a GWAS scenario would be statistically inappropriate given the imbalanced classification issue. The F1 score is the harmonic mean of SN and PPV, though due to space constraints we don’t really discuss it and we also recognize that finding an optimum balance between SN and PPV probably does not provide actionable guidance for functional follow-up. However, we provide the results necessary to determine which is the best-performing method if one wishes to maximize F1 score, of those methods evaluated here. We have added the following to lines 201-203: “To weight equally SN and PPV by choosing the method that yields the highest F1 score, or harmonic mean of SN and PPV, one can simply use the common suggestive p-value threshold of 1×10^{-5} and not bother with any of this.”
Minor question: 1. The question about method Selection (p7:272-278), please use different numeric ordinal levels to better distinguish the subtitle levels, such as 1) Categorized as a) annotation-based; b) pleiotropy-based; or c) eQTL-based; 2) Utilized GWAS summary statistics, as opposed to individual-level genotype data and so on.	We have used lowercase Roman numerals to list the methods criteria, letters to list the method categories, and retained the Arabic numerals to distinguish the different parts of the Methods sections.

2. Please check carefully the order of tables provided, such as p11: 468-469 (Additional details of the GWAS used to test the functional weighting methods are presented in Supplemental Table 1). The detailed information of GWAS used to test the functional weighting methods was provided in Supplemental Table 2.	This has been corrected. At the suggestion of another reviewer, the table describing the GWAS included in the study is now Table 2.
--	--

Reviewer #3	Responses
(Remarks to the Author): The authors present a comprehensive evaluation of functional weighting methods for identifying novel associations given GWAS summary statistics. The ongoing interest in GWAS and related down-stream analyses results in a multitude of potential applications of the study's outcome. For this reason this manuscript is certainly of interest for the readership of COMMSBIO.	
MAJOR COMMENTS:  Given the literature review has been conducted in 02/2020, I suggest to extend Table 1 by incorporating more recent methods (e.g. PLEIO [Lee et al. (2021)]). Furthermore, I recommend to update citation count and/or date of retrieval for the methods under investigation (>3,000 citations as of 10/2021). 	We have updated the citations to 7/11/2022 in Supplementary Table 1. As some methods have remained more popular than others, the statement of >3,000 citations (3,854 to be precise) is still accurate. We have revised line 238 from "...with >3,000 citations as of September 2021." To "...with >3,000 citations as of July 2022." As with any paper meant to synthesize previous work, a line must be drawn after which no additional published works will be considered. By systematically evaluating 18 methods across five traits, including a trait with no genome-wide significant variants, we stand to contribute substantively to the scientific literature upon this paper's publication. We will not be evaluating additional methods at this time.
 Selecting appropriate data for evaluating the weighting methods seems crucial. Therefore I suggest to include (an extended version of) Supplementary Table 2 into the main manuscript. 	We have moved Supplementary Table 2 into the main manuscript as Table 2; other tables have been renumbered to accommodate.
Furthermore, I recommend to discuss the following questions: (a) if sample size of two wave 1 studies differs considerably, how does this influence the outcome of the method evaluation?	Due to our use of publicly available summary statistics, we cannot answer this question by taking random subsamples of the GWAS1 participants, which would be ideal. However, we see from the three psychiatric traits that, despite

	roughly similar sample sizes for GWAS1, method performance is substantially affected less so by the number of study participants and more so the discovery power of that GWAS, which is a combination of the sample size, disease frequency, and trait-variant association magnitudes. We added to the discussion section, “Differences in the number of methods that were able to nominate any loci across the three psychiatric trait reiterate that trait heterogeneity, frequency, and variant association magnitude contribute in combination with study sample size to determine the minimum adequate GWAS to yield the first variant-trait associations³².” On lines 215-218.
(b) if increase of sample size from wave 1 to wave 2 differs considerably between two diseases, how does this influence the outcome of the method evaluation?	We cover this scenario in our study by including two blood cell traits whose wave 2 sample size is approximately 3X the wave 1 sample size, and three psychiatric traits whose wave 2 case sample size is 2.7-6.4X larger than the wave 1 case sample size. This scenario is also partially addressed by our sensitivity analysis comparing the functional weighting of wave 1 psychiatric traits against wave 3 GWAS of the same traits. In this case, with the even larger difference in sample sizes between wave 1 and the gold standard comparator, we found that, generally, PPV increased while SN decreased. We made no changes to the manuscript in response to this comment.
(c) if number of genotyped markers of two studies from wave 1 or wave 2 differs considerably, how does this influence the outcome of the method evaluation?	In a modern GWAS, one generally analyzes the set of genetic variants available after performing imputation using a suitable reference panel, rather than relying solely on the variants genotyped on the array. This is particularly relevant for studies conducted via meta-analysis, as most consortium-based studies are, and so we will respond to this question replacing “genotyped” with “imputed”. To answer, consider the case of evaluating methods using BPD and SCZ as model traits, as we did. Although the sample sizes of GWAS1 are roughly similar in value, BPD contains

	approximately twice the number of variants as SCZ (Table 2 in the revised submission). However, as shown in Supplemental Tables 3-4, PPVs are often, though not universally, higher when methods are applied to SCZ than to BPD. Consider also the comparison of MPV and WBC GWAS1 and GWAS2. The sample sizes and number of variants are approximately equal in both GWAS1 and both GWAS2. However, PPVs for WBC are generally higher than those for MPV; the publication describing GWAS1 for both these traits indicates that GWAS1 has just about saturated the genetic heritability due to common variation for MPV, but not for WBC. These two comparisons lead to the conclusion that the performance is more affected by how much of the underlying genetic architecture is captured by GWAS1 than the number of imputed variants under consideration, at least for contemporary imputation panels. We made no changes to the manuscript in response to this comment.
(d) if increase of number of genotyped markers from wave 1 to wave 2 differs considerably between two diseases, how does this influence the outcome of the method evaluation?	Changes in the number of genotyped or imputed markers from wave 1 to wave 2 are a typical expectation of successive GWAS waves. To artificially constrain the set of evaluated markers to the intersection of those genotyped on all arrays contributing to a given consortia GWAS1 meta-analysis would not reflect the reality of how GWAS2 are performed and would not contribute to a meaningful evaluation of functional weighting methods. We made no changes to the manuscript in response to this comment.
• As shown in Supplementary Table 4, for a given GWAS the number of input markers varies considerably between the different weighting methods (e.g. BPD: 2422487 (GWAS1 suggestive) vs. 188570 (fgwas), SCZ: 1242114 (GWAS1 suggestive) vs. 885859 (MTAG Using BPD)). How do these differences affect the outcome of the method evaluation? How would appropriate subsampling of markers affect the result?	We wanted to evaluate the methods as the authors intended them to be used, and method development does not necessarily consider the set of usable variants from other methods. We consider any stringent quality control or subsetting required by a given method a contribution to its performance relative to other methods. We made no changes to the manuscript in response to this comment.

 • I recommend to order the methods listed in Table 2 in accordance to their median rank. Furthermore, I suggest to extend Table 2 by variability estimates of PPV, e.g. generated based on the above-mentioned subsampling of markers. 	Table 2 has been reordered as suggested by the reviewer. Due to comments from another reviewer, it is now Table 3 in the resubmitted manuscript. As stated in our response to the reviewer's previous comment, we consider any stringent quality control or subsetting required by a given method a part of its performance relative to other methods.
MINOR COMMENTS:  • "[...] when focused on either high SN or high PPV, functional weighting GWAS methods boost statistical power [...]". This sentence partially contradicts with statements about methods with performances in Quadrant II of Figure 1 and 4 (low SN and high PPV), for which no boost in statistical power (=SN) has been observed. The following statement might be more appropriate: "Functional weighting GWAS methods might boost either SN or PPV where larger sample sizes are not feasible and the currently available GWAS has generated at least some genome-wide significant loci for the trait of interest." 	We thank the reviewer for this suggestion and we have implemented it. This change affects lines 283-288.
 • Please ensure concordance to the COMMSBIO Style and formatting guide by (a) renaming Citations to References, (b) adding a statement on Author contributions, (c) adding a statement on Competing interests, (d) labelling and citing Supplementary items as Supplementary Figure 1, Supplementary Table 1, Furthermore, please check whether "Neale, B. M. No Title. http://www.nealelab.is/uk-biobank/." corresponds to the Nature referencing style. 	We have made these changes throughout the manuscript as applicable.  Line 647 Lines 843-846 Lines 848-849 Throughout the manuscript To conform to the Nature referencing style, we have removed this webpage from the list of References and instead cited the URL within the text (e.g., line 356, line 366, line 621)
 • I recommend to check Supplementary Table 1, 2, 6, 7 for  correct cell contents (e.g. "ENSG locus calculated by...", "500kb locus calculated by...", "de Leeuw, et al. PLoS Comput Biol. 2016;11(40:e1004219"), consistent formatting (e.g. borders, alignment), explanations for abbreviations (ST1, 2, 3, 4, 5, 6, 7) (e.g. Supplementary Table 1: SCZ, BPD, MDD, MPV, WBC, HDL, PGC, UKBB, CHR, POS, BETA, SE, N, MAF, ProbeID, A1, A2, A1AF, P), 	We have made the recommended changes to the Supplementary Tables.

d) explanations for highlighting (e.g. Supplementary Table 6: highest ranks marked in bold font) and e) removal of unrelated information (e.g. Supplementary Table 2, B55:C60).	
---	--

REVIEWERS' COMMENTS:

Reviewer #1 (Remarks to the Author):

The authors have addressed my comments in a succinct and clear way.

Reviewer #2 (Remarks to the Author):

Thank you for the authors response. I have no more questions or comments. Only one minor comment was: In Table 1, the citation format of articles is not uniform. Most of them use author + journal + year, but some only cite author and year, while others only cite author.

Reviewer #3 (Remarks to the Author):

The authors present a comprehensive evaluation of functional weighting methods for identifying novel associations given GWAS summary statistics. The ongoing interest in GWAS and related down-stream analyses results in a multitude of potential applications of the study's outcome. For this reason this manuscript is certainly of interest for the readership of COMMSBIO.

RESPONSE BY REVIEWER #3: I thank the authors for responding to the entirety of the reviewers' comments. The revised version has been clearly improved in regard clarity and readability. Herewith I recommend the manuscript for publication.

MAJOR COMMENTS:

- Given the literature review has been conducted in 02/2020, I suggest to extend Table 1 by incorporating more recent methods (e.g. PLEIO [Lee et al. (2021)]). Furthermore, I recommend to update citation count and/or date of retrieval for the methods under investigation (>3,000 citations as of 10/2021).

RESPONSE FROM THE AUTHORS: We have updated the citations to 7/11/2022 in Supplementary Table 1. As some methods have remained more popular than others, the statement of >3,000 citations (3,854 to be precise) is still accurate. We have revised line 238 from "...with >3,000 citations as of September 2021." To "...with >3,000 citations as of July 2022."

As with any paper meant to synthesize previous work, a line must be drawn after which no additional published works will be considered. By systematically evaluating 18 methods across five traits, including a trait with no genome-wide significant variants, we stand to contribute substantively to the scientific literature upon this paper's publication. We will not be evaluating additional methods at this time.

RESPONSE BY REVIEWER #3: I thank the authors for updating the citation counts. Furthermore, I agree with their reasoning for not including additional methods.

- Selecting appropriate data for evaluating the weighting methods seems crucial. Therefore I suggest to include (an extended version of) Supplementary Table 2 into the main manuscript.

RESPONSE FROM THE AUTHORS: We have moved Supplementary Table 2 into the main manuscript as Table 2; other tables have been renumbered to accommodate.

RESPONSE BY REVIEWER #3: I thank the authors for reorganizing the tables.

- Furthermore, I recommend to discuss the following questions: (a) if sample size of two wave 1

studies differs considerably, how does this influence the outcome of the method evaluation?

RESPONSE FROM THE AUTHORS: Due to our use of publicly available summary statistics, we cannot answer this question by taking random subsamples of the GWAS1 participants, which would be ideal. However, we see from the three psychiatric traits that, despite roughly similar sample sizes for GWAS1, method performance is substantially affected less so by the number of study participants and more so the discovery power of that GWAS, which is a combination of the sample size, disease frequency, and trait-variant association magnitudes.

We added to the discussion section, "Differences in the number of methods that were able to nominate any loci across the three psychiatric trait reiterate that trait heterogeneity, frequency, and variant association magnitude contribute in combination with study sample size to determine the minimum adequate GWAS to yield the first variant-trait associations³²." On lines 215-218.

RESPONSE BY REVIEWER #3: I thank the authors for extending the discussion to include this important aspect.

- (b) if increase of sample size from wave 1 to wave 2 differs considerably between two diseases, how does this influence the outcome of the method evaluation?

RESPONSE FROM THE AUTHORS: We cover this scenario in our study by including two blood cell traits whose wave 2 sample size is approximately 3X the wave 1 sample size, and three psychiatric traits whose wave 2 case sample size is 2.7-6.4X larger than the wave 1 case sample size.

This scenario is also partially addressed by our sensitivity analysis comparing the functional weighting of wave 1 psychiatric traits against wave 3 GWAS of the same traits. In this case, with the even larger difference in sample sizes between wave 1 and the gold standard comparator, we found that, generally, PPV increased while SN decreased.

We made no changes to the manuscript in response to this comment.

RESPONSE BY REVIEWER #3: I thank the authors for the explanation and follow their reasoning for not extending the manuscript.

- (c) if number of genotyped markers of two studies from wave 1 or wave 2 differs considerably, how does this influence the outcome of the method evaluation?

RESPONSE FROM THE AUTHORS: In a modern GWAS, one generally analyzes the set of genetic variants available after performing imputation using a suitable reference panel, rather than relying solely on the variants genotyped on the array. This is particularly relevant for studies conducted via meta-analysis, as most consortium-based studies are, and so we will respond to this question replacing "genotyped" with "imputed".

To answer, consider the case of evaluating methods using BPD and SCZ as model traits, as we did. Although the sample sizes of GWAS1 are roughly similar in value, BPD contains approximately twice the number of variants as SCZ (Table 2 in the revised submission). However, as shown in Supplemental Tables 3-4, PPVs are often, though not universally, higher when methods are applied to SCZ than to BPD.

Consider also the comparison of MPV and WBC GWAS1 and GWAS2. The sample sizes and number of variants are approximately equal in both GWAS1 and both GWAS2. However, PPVs for WBC are generally higher than those for MPV; the publication describing GWAS1 for both these traits indicates that GWAS1 has just about saturated the genetic heritability due to common variation for MPV, but not for WBC.

These two comparisons lead to the conclusion that the performance is more affected by how much of the underlying genetic architecture is captured by GWAS1 than the number of imputed variants under consideration, at least for contemporary imputation panels.

We made no changes to the manuscript in response to this comment.

RESPONSE BY REVIEWER #3: I thank the authors for the explanation and follow their reasoning for not extending the manuscript. However, for completeness I suggest to complement the description of the incorporated datasets by a statement on whether observed or imputed genotypes have been studied, e.g. by extending Table 2.

- (d) if increase of number of genotyped markers from wave 1 to wave 2 differs considerably between two diseases, how does this influence the outcome of the method evaluation?

RESPONSE FROM THE AUTHORS: Changes in the number of genotyped or imputed markers from wave 1 to wave 2 are a typical expectation of successive GWAS waves. To artificially constrain the set of evaluated markers to the intersection of those genotyped on all arrays contributing to a given consortia GWAS1 meta-analysis would not reflect the reality of how GWAS2 are performed and would not contribute to a meaningful evaluation of functional weighting methods. We made no changes to the manuscript in response to this comment.

RESPONSE BY REVIEWER #3: I thank the authors for the explanation and agree on their statement on constraining the dimensionality of successive GWAS. However, for interpretability of the initial findings, I suggest to state the significance criteria (testing method, correction method and significance threshold) used within the individual studies. Again this could be done by extending Table 2.

- As shown in Supplementary Table 4, for a given GWAS the number of input markers varies considerably between the different weighting methods (e.g. BPD: 2422487 (GWAS1 suggestive) vs. 188570 (fgwas), SCZ: 1242114 (GWAS1 suggestive) vs. 885859 (MTAG Using BPD)). How do these differences affect the outcome of the method evaluation? How would appropriate subsampling of markers affect the result?

RESPONSE FROM THE AUTHORS: We wanted to evaluate the methods as the authors intended them to be used, and method development does not necessarily consider the set of usable variants from other methods. We consider any stringent quality control or subsetting required by a given method a contribution to its performance relative to other methods. We made no changes to the manuscript in response to this comment.

RESPONSE BY REVIEWER #3: I thank the authors for the explanation and follow their reasoning for not extending the manuscript.

- I recommend to order the methods listed in Table 2 in accordance to their median rank. Furthermore, I suggest to extend Table 2 by variability estimates of PPV, e.g. generated based on the above-mentioned subsampling of markers.

RESPONSE FROM THE AUTHORS: Table 2 has been reordered as suggested by the reviewer. Due to comments from another reviewer, it is now Table 3 in the resubmitted manuscript. As stated in our response to the reviewer's previous comment, we consider any stringent quality control or subsetting required by a given method a part of its performance relative to other methods.

RESPONSE BY REVIEWER #3: I thank the authors for editing Table 3. I follow their reasoning for omitting further extensions of the manuscript.

MINOR COMMENTS:

- "[...] when focused on either high SN or high PPV, functional weighting GWAS methods boost statistical power [...]". This sentence partially contradicts with statements about methods with performances in Quadrant II of Figure 1 and 4 (low SN and high PPV), for which no boost in statistical power (=SN) has been observed. The following statement might be more appropriate: "Functional weighting GWAS methods might boost either SN or PPV where larger sample sizes are not feasible and

the currently available GWAS has generated at least some genome-wide significant loci for the trait of interest."

RESPONSE FROM THE AUTHORS: We thank the reviewer for this suggestion and we have implemented it. This change affects lines 283-288.

RESPONSE BY REVIEWER #3: I thank the authors for editing the manuscript.

- Please ensure concordance to the COMMSBIO Style and formatting guide by (a) renaming Citations to References, (b) adding a statement on Author contributions, (c) adding a statement on Competing interests, (d) labelling and citing Supplementary items as Supplementary Figure 1, Supplementary Table 1, Furthermore, please check whether "Neale, B. M. No Title. <http://www.nealelab.is/uk-biobank/>." corresponds to the Nature referencing style.

RESPONSE FROM THE AUTHORS: We have made these changes throughout the manuscript as applicable.

a) Line 647

b) Lines 843-846

c) Lines 848-849

d) Throughout the manuscript

e) To conform to the Nature referencing style, we have removed this webpage from the list of References and instead cited the URL within the text (e.g., line 356, line 366, line 621)

RESPONSE BY REVIEWER #3: I thank the authors for editing the manuscript.

- I recommend to check Supplementary Table 1, 2, 6, 7 for correct cell contents (e.g. "ENSG locus calculated by...", "500kb locus calculated by...", "de Leeuw, et al. PLoS Comput Biol. 2016;11(40:e1004219)", consistent formatting (e.g. borders, alignment), explanations for abbreviations (e.g. Supplementary Table 1: SCZ, BPD, MDD, MPV, WBC, HDL, PGC, UKBB, CHR, POS, BETA, SE, N, MAF, ProbeID, A1, A2, A1AF, P), explanations for highlighting (e.g. Supplementary Table 6: highest ranks marked in bold font) and removal of unrelated information (e.g. Supplementary Table 2, B55:C60).

RESPONSE FROM THE AUTHORS: We have made the recommended changes to the Supplementary Tables.

RESPONSE BY REVIEWER #3: I thank the authors for editing the tables.

Thank you for the authors response. I have no more questions or comments. Only one minor comment was: In Table 1, the citation format of articles is not uniform. Most of them use author + journal + year, but some only cite author and year, while others only cite author.

Thank you for your review. We have corrected the citation format in Table 1.

RESPONSE BY REVIEWER #3: I thank the authors for the explanation and follow their reasoning for not extending the manuscript. However, for completeness I suggest to complement the description of the incorporated datasets by a statement on whether observed or imputed genotypes have been studied, e.g. by extending Table 2.

Thank you for your review and for your suggestion to provide more information about the genotype data used in our study. We have added a column to Table 2 indicating the imputation panel(s) used in each study. All GWAS used in the present study used imputed genotypes.

RESPONSE BY REVIEWER #3: I thank the authors for the explanation and agree on their statement on constraining the dimensionality of successive GWAS. However, for interpretability of the initial findings, I suggest to state the significance criteria (testing method, correction method and significance threshold) used within the individual studies. Again this could be done by extending Table 2.

Thank you for your comment. Most genome-wide association studies follow standard procedures for testing method, multiple testing correction, and significance threshold. What varies substantially from study to study is the process for determining statistically “independent” genome-wide significant loci. Thus, it would be helpful to more explicitly state how the original number of genome-wide significant variants or loci presented in each published study were determined, which contrasts with the number of genome-wide significant variants or loci used in our analysis. Although ostensibly generated using the same data, there are several important differences that were not immediately obvious from the previous versions of Table 2. First, not all data used in the original publication were available for public release, resulting in smaller sample sizes available for the present study than what was used in the published model trait GWAS. Second, some, but not all, model trait GWAS performed some sort of conditional analysis to identify statistically independent associations within a given locus. Third, the model trait GWAS cited were not identical in how they determined what findings met genome-wide significance with respect to the use discovery and replication samples.

To address these issues, we have added three columns to Table 2 presenting this information. The expanded Table 2 will make it easier to understand why the number of genome-wide significant variants or loci published in the original publications we cite do not exactly correspond to the number used in the present analysis.

REVIEWERS' COMMENTS:

Reviewer #1 (Remarks to the Author):

No additional comments.

Reviewer #2 (Remarks to the Author):

The authors tried to identify suitable method(s) for improving GWAS statistical power to uncover novel loci from available 18 methods which have compared the ability of existing methods to uncover novel loci, although no method achieved both high PPV and FN. According to the response of authors, I have no more questions or comments.

Reviewer #3 (Remarks to the Author):

The authors present a comprehensive evaluation of functional weighting methods for identifying novel associations given GWAS summary statistics. The ongoing interest in GWAS and related down-stream analyses results in a multitude of potential applications of the study's outcome. For this reason this manuscript is certainly of interest for the readership of COMMSBIO.

RESPONSE BY REVIEWER #3: I thank the authors for responding to the entirety of the reviewers' comments. The revised version has been clearly improved in regard clarity and readability. Herewith I recommend the manuscript for publication.

MAJOR COMMENTS:

- (c) if number of genotyped markers of two studies from wave 1 or wave 2 differs considerably, how does this influence the outcome of the method evaluation?

RESPONSE FROM THE AUTHORS: Thank you for your review and for your suggestion to provide more information about the genotype data used in our study. We have added a column to Table 2 indicating the imputation panel(s) used in each study. All GWAS used in the present study used imputed genotypes.

RESPONSE BY REVIEWER #3: I thank the authors for updating Table 2.

- (d) if increase of number of genotyped markers from wave 1 to wave 2 differs considerably between two diseases, how does this influence the outcome of the method evaluation?

RESPONSE FROM THE AUTHORS: Thank you for your comment. Most genome-wide association studies follow standard procedures for testing method, multiple testing correction, and significance threshold. What varies substantially from study to study is the process for determining statistically "independent" genome-wide significant loci. Thus, it would be helpful to more explicitly state how the original number of genome-wide significant variants or loci presented in each published study were determined, which contrasts with the number of genome-wide significant variants or loci used in our analysis. Although ostensibly generated using the same data, there are several important differences that were not immediately obvious from the previous versions of Table 2. First, not all data used in the original publication were available for public release, resulting in smaller sample sizes available for the present study than what was used in the published model trait GWAS. Second, some, but not all, model trait GWAS performed some sort of conditional analysis to identify statistically independent associations within a given locus. Third, the model trait GWAS cited were not identical in how they determined what findings met genome-wide significance with respect to the use discovery and

replication samples.

To address these issues, we have added three columns to Table 2 presenting this information. The expanded Table 2 will make it easier to understand why the number of genome-wide significant variants or loci published in the original publications we cite do not exactly correspond to the number used in the present analysis.

RESPONSE BY REVIEWER #3: I thank the authors for extending Table 2.

ADDITIONAL COMMENTS FROM THE AUTHORS:

In the course of making these revisions, we discovered that we had unintentionally used the summary statistics for the trait "platelet distribution width" instead of "mean platelet volume" for GWAS2 of that trait; the trait IDs differ by a single digit. We have rerun all mean platelet volume comparisons and found that our conclusions have not changed.

We identified a minor error in calculating the Bonferroni correction for the MAGMA-based evaluations of the method JEPEG; this has also been corrected and has not changed our conclusions.

RESPONSE BY REVIEWER #3: I thank the authors for accounting for the discovered errors as well as double-checking the derived conclusions.

ADDITIONAL COMMENTS FROM THE AUTHORS:

We identified two minor errors in the supplementary figures. We include a revised Supplementary Figure 1 showing the sensitivity vs positive predictive value for the three psychiatric traits and a revised Supplementary Figure 4b showing the UpSet plot for bipolar disorder. We have also revised Supplementary Table 6, formerly 7, to reflect the correct distribution of method combinations and the text describing these results in lines 164-175 of the tracked changes manuscript.

RESPONSE BY REVIEWER #3: I thank the authors for accounting for the discovered errors updating the respective plots and table. However, I suggest to correct the caption of Sheet1 of Supplementary Data 9 ("Supplementary Data 9a: Data for Schizophrenia Upset Plot Supplementary Figure 4a").

MINOR COMMENTS:

- I suggest to review all changes made to the manuscript for mistakes, e.g. duplicated "that" in line 91-92, duplicated "Supplementary Data" in line 90, 96, 120, 124, 127-128, 140, 142 and 146, missing whitespace in line 192.
- When examining functional weighting methods the authors use "PPV > 50%" as a quality measure. I suggest to use "PPV ≤ 50%" instead of "PPV < 50%" for methods not fulfilling the criterion.
- Throughout the manuscript the authors mention 17 (title, abstract, results, methods) but also 18 methods including a threshold-based approach (introduction, results, methods). I suggest to use a consistent counting of the methods under consideration.

Point-by-point responses:

We thank reviewer 3 for their continued attention to detail and thoughtful consideration of the journal editorial staff. We make the following point-by-point responses:

- I suggest to review all changes made to the manuscript for mistakes, e.g. duplicated “that” in line 91–92, duplicated “Supplementary Data” in line 90, 96, 120, 124, 127–128, 140, 142 and 146, missing whitespace in line 192.

We thank the reviewer for these comments. We have addressed these issues.

- When examining functional weighting methods the authors use "PPV > 50%" as a quality measure. I suggest to use "PPV ≤ 50%" instead of "PPV < 50%" for methods not fulfilling the criterion.

In the manuscript we use PPV threshold as descriptive characterizations of our results. We are not advocating for the field to adopt a threshold of $\geq 50\%$ PPV as a minimum quality criterion for functional weighting methods used singly or as an ensemble. We use 50% as a demarcating value for two reasons of convenience: it divides the scatterplots of sensitivity vs positive predictive value into four equal quadrants, and it is colloquially familiar as the performance of a fair coin toss, which gives readers an easy point of reference. When describing Figure 1 in the manuscript text, we use $PPV > 0.50$, as this describes the location of what is indisputably Quadrant I of the plot. Boundary conditions are always tricky. When describing the consistency of true associations nominated across methods, we use $PPV \geq 50\%$ as this describes our results presented in the table.

We have made no changes to the manuscript in response to this comment.

- Throughout the manuscript the authors mention 17 (title, abstract, results, methods) but also 18 methods including a threshold-based approach (introduction, results, methods). I suggest to use a consistent counting of the methods under consideration.

We agree that this is a potential point of confusion. The perceived inconsistency of methods evaluated results from the definition of “functional weighting method”. We maintain that describing 17 approaches as “functional weighting” is important to quickly and concisely inform our readership of the nature of these 17 approaches. The 18th method simply uses the suggestive p-value threshold of $p < 1 \times 10^{-5}$ to nominate variants, which is not “functional weighting”. It is not possible to accurately describe our approach of evaluating “17 published functional weighting methods and a suggestive p-value threshold as an 18th method” in the title and abstract while meeting the strict word limits. We have therefore chosen to emphasize the 17 functional weighting methods for the sake of both brevity and accuracy. To ensure consistency in making this distinction, revised line 59-61 in the introduction to read

“To identify suitable method(s) for improving GWAS statistical power to uncover novel loci, we performed the largest, most comprehensive evaluation of published functional weighting methods to date: 17 functional weighting methods, and an unweighted suggestive p-value threshold, applied to multiple waves of GWAS for five diseases and traits.”

We also revised subheading 9 of the Methods section (line 604-605) to read
“Application of 17 Functional Weighting Methods and A Suggestive Threshold To Model Traits”

Further, I found the following minor formatting issues that need to be fixed:

- Each affiliation must include the institution, city and country.

We have double checked all affiliations, which comply with this requirement.

- In the Methods section, please use only 1 level of subheadings.

Due to the complexity of the manuscript methods, we maintain that the additional subheadings provide invaluable clarity to the unfamiliar reader and advocate for an exception to be made. We leave this for the journal editorial staff to arbitrate.

- Please retitle Supplementary Note 2 and 3 to Supplementary Software 1-2 and cite these the Code Availability statement.

It does not seem that our code constitutes software. We think our scripts are well-distributed in the form of Supplementary Notes. These Notes are cited in the Code Availability statement.

- Shading needs to be removed from Table 3

We have done so in the revised document.

- Figure legends/titles should be removed from the individual figure files.

We have done so in the revised document.